# A modified impulse-response representation of the global near-surface air temperature and atmospheric concentration response to carbon dioxide emissions

Richard J. Millar[1,2,3], Zebedee. R. Nicholls[1,4,5], Pierre Friedlingstein[3], and Myles R. Allen[1,2,6]

[1]Department of Physics, University of Oxford, Oxford, UK
[2]Oxford Martin Net Zero Carbon Investment Initiative, Oxford Martin School, University of Oxford, Oxford, UK
[3]Department of Mathematics, University of Exeter, Exeter, UK
[4]Australian-German Climate & Energy College, The University of Melbourne, Parkville, Victoria, Australia
[5]Department of Earth Sciences, The University of Melbourne, Parkville, Victoria, Australia
[6]Environmental Change Institute, University of Oxford, Oxford, UK

*Correspondence to:* Richard J. Millar (richard.millar@physics.ox.ac.uk)

**Abstract.** Projections of the response to anthropogenic emission scenarios, evaluation of some greenhouse gas metrics, and estimates of the social cost of carbon often require a simple model that links emissions of carbon dioxide ($CO_2$) to atmospheric concentrations and global temperature changes. An essential requirement of such a model is to reproduce typical global surface temperature and atmospheric $CO_2$ responses displayed by more complex Earth System Models (ESMs) under a range of emissions scenarios, as well as an ability to sample the range of ESM response in a transparent, accessible and reproducible form. Here we adapt the simple model of the Intergovernmental Panel on Climate Change 5th Assessment Report (IPCC-AR5) to explicitly represent the state-dependence of the $CO_2$ airborne fraction. Our adapted model (FAIR) reproduces the range of behaviour shown in full and intermediate complexity ESMs under several idealised carbon-pulse and exponential concentration increase experiments. We find that the inclusion of a linear increase in 100-year integrated airborne fraction with cumulative carbon uptake and global temperature change substantially improves the representation of the response of the climate system to $CO_2$ on a range of timescales and under a range of experimental designs.

## 1 Introduction

In the long term, future climate changes will largely be determined by future cumulative $CO_2$ emissions (Matthews et al., 2009; Allen et al., 2009; Meinshausen et al., 2009), but the timing and magnitude of emissions are uncertain and a strong function of future climate policy (Van Vuuren et al., 2011). Linking specific $CO_2$ emission scenarios to future transient climate change requires a model of the interacting climate-carbon-cycle system. Comprehensive ESMs explicitly simulate the physical processes that govern the coupled evolution of atmospheric $CO_2$ concentrations and the associated climate response (Friedlingstein et al., 2006). However, the use of such models may be computationally intensive, require large ensembles to distinguish climate change signals from internal variability, and often require substantial post-processsing of output. Therefore ESMs are often only run for a few representative future emission scenarios (Taylor et al., 2012). For analysis of arbitrary emissions scenarios,

as required for the integrated assessment of climate policy, a computationally efficient representation of the Earth system is needed (Marten, 2011), particularly in order to sample Earth system response uncertainty comprehensively in probabilistic frameworks.

Simplified representations of the coupled climate-carbon-cycle system take many forms (Hof et al., 2012). A key test for simplified ESMs is whether they correctly capture the physics of the co-evolution of atmospheric $CO_2$ concentrations and global mean temperature under both idealised settings and under possible projections of future emissions scenarios. Following a $CO_2$ pulse emission of 100GtC in present-day climate conditions, ESMs (and Earth System Models of Intermediate Complexity – EMICs) display a rapid-draw down of $CO_2$ with the concentration anomaly reduced by approximately 40% from peak after 20 years and by 60% after 100 years, followed by a much slower decay of concentrations which leaves approximately 25% of peak concentration anomaly remaining after 1000 years (Joos et al., 2013). The speed and shape of this decay is dependent on both the background climate state and the size of the pulse, but a substantial fraction of the emission is simulated to remain in the atmosphere after 1000 years in all cases. The effect of this longevity of fossil carbon in the atmosphere, combined with the gradual "recalcitrant" thermal adjustment of the climate system (Held et al., 2010), is to induce a global mean surface temperature response to a pulse emission of $CO_2$ characterised by a rapid warming, over approximately a decade, to a plateau value of global mean surface temperature anomaly (Joos et al., 2013). Warming does not noticeably decrease from this value over the following several hundred years, indicating that, short of artificial $CO_2$ removal (CDR) or active solar geoengineering, $CO_2$-induced warming is essentially permanent on human-relevant timescales.

The correct representation of the temporal evolution of the warming response to a pulse emission is required for computationally-simple climate-carbon-cycle models. Aside from the simple climate-carbon-cycle models analysed in Joos et al. (2013), many simple models, including some used in integrated assessment models (IAMs - e.g see Nordhaus (2010)), have not explicitly been evaluated in terms of their pulse-response behaviour, and it remains unclear how well the physical dependences of the pulse-response are represented in such models. The social cost of carbon is conventionally calculated by applying a pulse emission of a specified magnitude in near to present-day conditions as a perturbation on top of a specified future emissions scenario (NAS, 2016). As calculating the social cost of carbon is a key element of many cost-benefit analyses of climate change policy, simple climate-carbon-cycle models used in IAMs should aim to reproduce the pulse-response dependencies on pulse size and background state that have been highlighted in ESMs and EMICs (Joos et al., 2013; Herrington and Zickfeld, 2014).

A second important feature of ESMs is the increase in airborne fraction (the fraction of emitted $CO_2$ that remains in the atmosphere after a specified period) over time in scenarios involving substantial emissions or warming (Friedlingstein et al., 2006). An emergent feature of the CMIP5 full-complexity ESMs appears to be that this increase in airborne fraction approximately cancels the logarithmic relationship between $CO_2$ concentrations and radiative forcing, yielding an approximately linear relationship between cumulative $CO_2$ emissions and $CO_2$-induced warming (Matthews et al., 2009; Gillett et al., 2013). This relationship has given rise to the concept of an all-time cumulative 'carbon budget' to restrict warming to a certain level (Rogelj et al., 2016), which has quickly become an important tool in evaluating the required energy-system transitions that are needed to limit warming to below particular thresholds (Gignac and Matthews, 2015; Van Vuuren et al., 2016), as well as the climate implications of the existing capital stock (Davis and Socolow, 2014; Pfeiffer et al., 2016). As simple climate-carbon-cycle

models are often used to compute particular carbon budgets in integrated assessment scenarios (e.g. the MAGICC model as used in Meinshausen et al. (2009)), the ability to reproduce the approximate linearity of the relationship between warming and cumulative emissions is a desirable property.

Representing climate response uncertainty is also a crucial factor in the integrated assessment of climate policies. Despite significant advances in climate system understanding, non-negligible uncertainties remain in the response of the coupled climate-carbon-cycle system to emissions of $CO_2$ (Gillett et al., 2013) implying that climate policies have to be constructed and assessed in the light of this continued uncertainty (Millar et al., 2016). Integrated assessment activities require a representation of the physical climate system that can transparently and simply sample physically-consistent modes of climate response uncertainty, partly to assess the possibility of extreme and highly costly responses within the Earth system (often called "fat-tailed" outcomes) (Weitzman, 2011).

In this paper we show that although the impulse-response functions provided for the calculation of multi-gas equivalence metrics in IPCC-AR5 (Myhre et al., 2013) provide a simple and easy to use climate-carbon-cycle model, this model is insufficient to fully capture the emergent responses of the coupled climate-carbon-cycle system. Such a state-independent impulse-response model cannot simultaneously reproduce the relationship between emissions, concentrations and temperatures seen over the historical period and in the projected response over the $21^{st}$ century to both high-emissions and mitigation scenarios as simulated by ESMs and EMICs. Indeed, such a model formalism would inherently fail to capture the dependence of the evolution of the airborne fraction following a pulse emission on both the background state of the climate and pulse size, as simulated by ESMs in Joos et al. (2013). We therefore propose a simple extension of the standard IPCC-AR5 impulse-response model, coupling the carbon-cycle to the thermal response and to cumulative carbon uptake by terrestrial and marine sinks in order to reproduce the behaviour of the ESMs under a variety of idealised experiments and future emissions scenarios.

Section 2 describes the formalism of the models that we contrast throughout this paper and describes the methodological details of the experiments that we use to analyse the responses of these models. We show and discuss the results of these model validation experiments in section 3, beginning, in section 3.1, with why a state-dependence modification to the IPCC-AR5 carbon-cycle impulse-response function is required, motivating the modified 'Finite Amplitude Impulse-Response Model' (FAIR) described in section 2. Section 3.2 then evaluates the ability of FAIR and the unmodified IPCC-AR5 impulse-response models to replicate the dependencies of the response to a pulse-emission on background conditions and pulse size shown in ESMs and EMICs. Section 3.3 evaluates the models' behaviour under a set of idealised experiments in which $CO_2$ concentrations are increased by a fixed percentage each year starting from pre-industrial values. Section 3.4 discusses uncertainty in FAIR and how probabilistic assessments of climate response to $CO_2$ emissions could be made using the model. Section 4 provides a concluding summary and discussion.

## 2 Model description and methods

### 2.1 The IPCC AR5 Impulse-Response (AR5-IR) model

The IPCC-AR5 proposed an idealised simple climate-carbon-cycle model for metric calculations, incorporating a "2-box" or "2-time-constant" model of the temperature response to radiative forcing with a "4-time-constant" impulse-response model of the $CO_2$ concentration response to emissions (Myhre et al., 2013). This model represents the evolution of atmospheric $CO_2$ by partitioning emissions of anthropogenic $CO_2$ between four different reservoirs (all of which are empty in pre-industrial equilibrium) of atmospheric carbon anomaly that each decay with a fixed time constant. Four carbon pools are determined to be sufficient to empirically represent the response of atmospheric $CO_2$ concentration anomalies following a pulse emission of 100GtC, above a specified background concentration of 389ppm, over the 1000 years following the pulse (Joos et al., 2013). These carbon pools do not directly correspond to individual physical processes and instead represent the combined effect of several carbon-cycle mechanisms, however, processes that are guiding analogues to the timescale of the pool decays are summarised in table 1. The evolution of the carbon concentration anomaly in each pool, $R_i$, is given as,

$$\frac{dR_i}{dt} = a_i E - \frac{R_i}{\tau_i} \quad ; \quad i = 1 - 4 \tag{1}$$

where $E$ is the annual $CO_2$ emissions, in units of ppm/year (1 ppm = 2.12GtC), $a_i$ is the fraction of carbon emissions entering each reservoir and $\tau_i$ the decay time constant for that pool. Coefficients $a_i$, and $\tau_i$ are as given in AR5 Chapter 8, tables 8.SM.9 and 8.SM.10 (Myhre et al., 2013), except for $\tau_0$ which is here given a finite value and not set to infinity (all results presented in this paper are insensitive to this choice). Atmospheric $CO_2$ concentrations are given by $C = C_0 + \sum_i R_i$, and radiative forcing by:

$$F = \frac{F_{2X}}{\ln(2)} \ln\left(\frac{C}{C_0}\right) + F_{ext} \quad , \tag{2}$$

where $C_0$ is the pre-industrial $CO_2$ concentration, $F_{2X}$ the forcing due to $CO_2$ doubling ($F_{2\times}$=3.74Wm$^{-2}$), and $F_{ext}$ the non-$CO_2$ forcing. Global mean surface temperature anomalies ($T$) are computed as the sum of a two components ($T_j$) representing the contributions to global mean surface temperature anomalies controlled by the equilibration timescale of the upper/deep ocean respectively:

$$\frac{dT_j}{dt} = \frac{q_j F - T_j}{d_j} \quad ; \quad T = \sum_j T_j \quad ; \quad j = 1, 2. \tag{3}$$

The two thermal response timescales, $d_j$, have been ordered to run from longest to slowest, as with the carbon-cycle response timescales, and are chosen to match the multi-model means of Geoffroy et al. (2013). By considering the analytic solutions of equation 3 under an instantaneous doubling of $CO_2$ concentrations and a 1%/yr increase in $CO_2$ concentrations, $q_j$ can be related to the Equilibrium Climate Sensitivity (ECS[1]) and Transient Climate Response (TCR[2]) via the expressions,

$$ECS = F_{2\times}(q_1 + q_2), \tag{4}$$

---

[1]The global mean warming resulting from an instantaneous doubling of pre-industrial $CO_2$ concentrations after allowing the climate system to reach a new equilibrium state.

[2]The global mean warming at the time of doubled $CO_2$ concentrations following a 1%/yr increase from pre-industrial values

| Parameter | Value - AR5-IR | Value - PI-IR | Value - FAIR | Guiding analogues |
|---|---|---|---|---|
| $a_0$ | 0.2173 | 0.1545 | 0.2173 | Geological re-absorption |
| $a_1$ | 0.2240 | 0.1924 | 0.2240 | Deep ocean invasion / equilibration |
| $a_2$ | 0.2824 | 0.2424 | 0.2824 | Biospheric uptake / ocean thermocline invasion |
| $a_3$ | 0.2763 | 0.4108 | 0.2763 | Rapid biospheric uptake / ocean mixed-layer invasion |
| $\tau_0$ (yr) | $1\times10^6$ | $1\times10^6$ | $1\times10^6$ | Geological re-absorption |
| $\tau_1$ (yr) | 394.4 | 276.7 | 394.4 | Deep ocean invasion/equilibration |
| $\tau_2$ (yr) | 36.54 | 30.75 | 36.54 | Biospheric uptake / ocean thermocline invasion |
| $\tau_3$ (yr) | 4.304 | 4.459 | 4.304 | Rapid biospheric uptake / ocean mixed-layer invasion |
| $q_1$ ($KW^{-1}m^2$) | 0.33 | 0.33 | 0.33 | Thermal equilibration of deep ocean |
| $q_2$ ($KW^{-1}m^2$) | 0.41 | 0.41 | 0.41 | Thermal adjustment of upper ocean |
| $d_1$ (yr) | 239.0 | 239.0 | 239.0 | Thermal equilibration of deep ocean |
| $d_2$ (yr) | 4.1 | 4.1 | 4.1 | Thermal adjustment of upper ocean |
| $r_0$ (yr) | - | - | 32.40 | Pre-industrial iIRF$_{100}$ |
| $r_C$ (yr/GtC) | - | - | 0.019 | Increase in iIRF$_{100}$ with cumulative carbon uptake |
| $r_T$ (yr/K) | - | - | 4.165 | Increase in iIRF$_{100}$ with warming |

**Table 1.** Default parameter values for the simple impulse-response climate-carbon-cycle models used in this paper.

and

$$\text{TCR} = F_{2\times}\left(q_1\left(1 - \frac{d_1}{70}\left(1 - \exp\left[-\frac{70}{d_1}\right]\right)\right) + q_2\left(1 - \frac{d_2}{70}\left(1 - \exp\left[-\frac{70}{d_2}\right]\right)\right)\right), \tag{5}$$

Equations 4 and 5 can be inverted to give expressions for $q_j$ in terms of ECS and TCR assuming response timescales ($d_j$) as given in table 1 (Millar et al., 2015). We choose default values for $q_j$ corresponding to TCR=1.6K and ECS=2.75K ($q_1 = $ 5  $0.33KW^{-1}m^2$ and $q_2 = 0.41KW^{-1}m^2$), indicative of a typical mid-range climate response to radiative forcing in ESMs (Flato et al., 2013).

We use two versions of the AR5-IR model in this paper, calibrated to the present-day (AR5-IR) and pre-industrial (PI-IR) climate response to a pulse emission respectively. The AR5-IR model is used for the calculation of absolute Global Temperature Potentials (aGTPs) in IPCC-AR5 and has carbon-cycle coefficients that best represent the ESM simulated evolution of a 10  100GtC pulse emission under approximately present-day conditions. The PI-IR model uses an alternative set of coefficients that are selected to represent the evolution of a 100GtC pulse emission in pre-industrial conditions. These parameters are derived from a fit to the multi-model mean of the ensemble of ESMs and EMICs from Joos et al. (2013) (see table 1 for parameter values).

## 2.2 A Finite Amplitude Impulse Response (FAIR) model

15  In the AR5-IR and PI-IR models the carbon-cycle response to a pulse emission is not explicitly affected by rising temperature or $CO_2$ accumulation and hence these models only represent the specific response to a particular perturbation scenario. In more

comprehensive models, ocean uptake efficiency declines with accumulated $CO_2$ in ocean sinks (Revelle and Suess, 1957) and uptake of carbon into both terrestrial and marine sinks are reduced by warming (Friedlingstein et al., 2006).

In an attempt to capture some of these dynamics within the simple impulse-response model structure, we attempt a minimal modification of the AR5-IR model to allow it to mimic the behaviour of ESMs/EMICs in response to finite-amplitude $CO_2$ injections, which we call a Finite Amplitude Impulse-Response (FAIR) model. To introduce a state-dependent carbon uptake as simply as possible, we apply a single scaling factor, $\alpha$, to all four of the time-constants in the carbon-cycle of the AR5-IR model, such that the $CO_2$ concentrations in the 4 "carbon reservoirs" are updated thus:

$$\frac{\mathrm{d}R_i}{\mathrm{d}t} = a_i E - \frac{R_i}{\alpha \tau_i} \quad ; \quad i = 1 - 4 \tag{6}$$

To identify a suitable state-dependence, we focus on parameterising variations in the 100-year integrated impulse response function, $\text{iIRF}_{100}$. A focus on the integrated impulse response (average airborne fraction over a period of time, multiplied by the length of time period), as opposed to the airborne fraction at a particular point in time, is more closely related to the impact of $CO_2$ emissions on the global energy budget, and also to other metrics such as Global Warming Potential (GWP). With other coefficients fixed, $\text{iIRF}_{100}$ is a monotonic (but non-linear) function of $\alpha$:

$$\text{iIRF}_{100} = \sum_i \alpha a_i \tau_i \left[ 1 - \exp\left( \frac{-100}{\alpha \tau_i} \right) \right]. \tag{7}$$

As equation 7 is derived using the approximation that $\alpha$ is independent of time, the right hand side of equation 7 is only exactly equivalent to the $\text{iIRF}_{100}$ for infinitesimal pulse emission perturbations from a constant background climate state (in which the approximation of time-independent $\alpha$ becomes exact). We assume $\text{iIRF}_{100}$ is a function of the accumulated perturbation carbon stock in the land and ocean (equivalent to the amount of emitted carbon that no longer resides in the atmosphere), $C_{\text{acc}} = \sum_t E - (C - C_0)$, and of the global mean temperature anomaly from pre-industrial conditions, $T$. A simple linear relationship appears to give an adequate approximation to the behaviour of ESMs and EMICs (as will be shown subsequently in section 3):

$$\text{iIRF}_{100} = r_0 + r_C C_{\text{acc}} + r_T T. \tag{8}$$

At each time-step we first compute the required $\text{iIRF}_{100}$ using $C_{\text{acc}}$ and $T$ from the previous time-step (equation 8). We then numerically solve equation 7 for the compatible value for $\alpha$, which is then in turn used to update the carbon pool concentrations (equation 6). The total radiative forcing is then computed with equation 2, before changes in global mean temperature are computed with equation 3.

Values of $r_0$=32.4 years, $r_C$=0.019 years/GtC, $r_T$=4.165 years/K, ECS=2.75K and TCR=1.6K are here used as model default parameters[3]. We choose these parameters to approximately replicate the relationship between warming-driven outgassing of carbon in the bulk of CMIP5 ESMs (see section 3.3), whilst also diagnosing near-observed values of present-day $CO_2$ emissions to achieve present-day concentrations. The values given here as default parameters are intended to be taken only as approximate

---

[3]The carbon-cycle decay timescale scaling factor $\alpha$ is not restricted to be $> 1$. With default parameters given in table 1 the value of $\alpha$ in the pre-industrial state is 0.11.

CMIP5-representative values that capture important carbon-cycle dynamics in ESMs. These values have not been explicitly optimised to any particular goal and can be tuned (along with the other model parameters) to reproduce specific aspects of individual ESM/EMIC behaviour (e.g. see figure 4). Best-estimate values for the FAIR parameters will depend on exactly what feature of ESM behaviour is the desired target for the optimisation.

Values of iIRF$_{100}$ larger than 100 years correspond to a net carbon source to the atmosphere in response to a perturbation, and, as perturbations to the carbon stock in the atmosphere would grow indefinitely, makes the model unstable. In this regime there is no solution for $\alpha$, so we set iIRF$_{100}$ to a maximum value of 96.6 years, corresponding, with the parameters as given in table 1, to $\alpha$=100. This physically corresponds to a near-absence of carbon sinks in the Earth system following a very large injection, with very slow rates of decay of atmospheric concentrations. This limit is only reached after 2250 in RCP8.5 (Riahi
et al., 2011) of the scenarios considered in this paper and is unimportant for the results presented in section 3.

## 2.3  Experimental set-up

In this section we describe the features of several experimental protocols that have been used to examine coupled climate-carbon-cycle feedbacks in ESMs and EMICs. These experiments, conducted with the AR5-IR, PI-IR and FAIR models, form the core of our analysis of these models in section 3.

### 2.3.1  Pulse-response experiments

Joos et al. (2013) documented the response of an ensemble of ESMs and EMICs to pulses of various sizes and under various background conditions (black lines in figure 3). In the PD100 experiment (100GtC pulse in approximately present-day background conditions), background emissions are diagnosed that stabilise $CO_2$ concentrations at 389ppm (after rising as historically observed). In a second experiment, a 100GtC pulse is added to these diagnosed background emissions in the year that
$CO_2$ concentrations reach 389ppm and the resulting $CO_2$ concentration and temperature evolutions are compared to the case without the pulse emission to isolate the response to the pulse emission alone. Experiments were also conducted for a pulse of 100GtC and 5000GtC in pre-industrial background conditions (PI100 and PI5000 respectively) for a smaller sub-set of models.

We simulate these experiments with the impulse-response climate-carbon-cycle models by following the experimental protocol exactly as described in Joos et al. (2013). Emissions are derived consistent with the background concentration profile
using an inversion of the carbon-cycle equations for the AR5-IR and PI-IR models (equation 1), and the FAIR model (equation 6). A declining but non-zero low level of diagnosed emissions are required to stabilise atmospheric concentrations at the 389 ppm level for all of the models considered.

As well as investigating the response of the default FAIR parameters in these pulse-response experiments, we also investigate how parameter perturbations could allow FAIR to span the range of responses observed in the PD100 and PI100 experiments
for the individual models of the Joos et al. (2013) ensemble. We fit the FAIR parameters to the individual model responses in a two-step process. First, the carbon-cycle parameters ($a_i$, $r_0$, $r_T$ and $r_C$) of the FAIR model are optimised to minimise the total combined residual sum of squares of the FAIR fit to the Joos et al. (2013) multi-model mean airborne fraction across both the PD100 and PI100 experiments. As a constraint on this fit, we fix the ratio between the $r_T$ and $r_C$ parameters at

the value of this for the default parameters given in table 1. This is both to reduce the number of free parameters in the fitting process (the model is underconstrained as pulse-response experiments don't distinguish between temperature-induced and $CO_2$ uptake-induced carbon-cycle feedbacks), and because the representation of the temperature-induced carbon-cycle feedbacks in the 'radiatively-coupled' prescribed concentration increase experiment (see section 2.3.2) is more sensitive to parameter
perturbations that change this ratio than to perturbations that don't alter it (not shown).

After fitting the multi-model mean as described above, we then fit the responses for individual models by minimising the combined PD100 and PI100 residual sum of squares whilst allowing only the $r_0$, $r_T$ and $r_C$ parameters to vary from the model parameters found in the multi-model mean fit (the ratio between $r_T$ and $r_C$ is again fixed at the value for the default parameters). The timeseries of change in global mean surface temperature due to the pulse emission are taken as simulated by
the individual models when conducting both stages of these fits.

We also consider the response of the FAIR and AR5-IR models under the idealised pulse experiments of Herrington and Zickfeld (2014). Herrington and Zickfeld (2014) conducted several experiments with the UVic Earth System Model of intermediate complexity (Weaver et al., 2001). We here emulate the PULSE experiments of Herrington and Zickfeld (2014) by integrating the FAIR and AR5-IR models with historical fossil fuel and land-use $CO_2$ emissions together with estimates of the
historical non-$CO_2$ radiative forcing, both backed-out from historical concentrations using the MAGICC model (Meinshausen et al., 2011). Pulse emissions of various sizes were then applied over a two-year period from 2008 in order to restrict total all time cumulative emissions to specified totals (see Herrington and Zickfeld (2014) for details). Non-$CO_2$ forcings are held constant at 2008 levels after following RCP8.5 trajectories for 2005-2008.

### 2.3.2 Exponential $CO_2$ increase experiments

To explore the response to sustained emissions, rather than an emission pulse, we consider the experiments of Gregory et al. (2009) and Arora et al. (2013), in which ESMs are subjected to specified rates of increase in $CO_2$ concentrations. Concentrations were increased from pre-industrial values at $0.5\%yr^{-1}$, $1\%yr^{-1}$ and $2\%yr^{-1}$ respectively and consistent emissions were diagnosed for different configurations of the ESMs: a "biogeochemically-coupled" experiment, where the carbon-cycle is only allowed to respond to the direct effect of increasing $CO_2$ concentrations and not to the resultant warming; a "radiatively-
coupled" experiment in which the climate system is allowed to respond to the radiative forcing of $CO_2$ but the carbon-cycle is only allowed to respond to the simulated warming and not to increasing $CO_2$; and a "fully-coupled" experiment in which the carbon-cycle is allowed to respond to both warming and increased $CO_2$. Such idealised scenarios can be highly informative regarding the physical drivers of carbon-cycle feedbacks under increasing emissions.

Within the FAIR framework we recreate the "biogeochemically-coupled" experiment by setting $r_T$ =0, and approximate the
"radiatively-coupled" experiment by evaluating the difference between the "fully-coupled" and "biogeochemically-coupled" experiments (a net out-gassing of carbon, the simulated response to the "radiatively-coupled" experiment in the ESMs, cannot be directly simulated in impulse-response models where a pulse emission of carbon always decays over time). Although Gregory et al. (2009) found that the relationship between the experiments was not simply a linear summation at high $CO_2$ concentrations, this serves as an adequate approximation for our purposes since our objective is the correct representation of

aggregate feedbacks from different effects in the FAIR model as opposed to a more complex linear and non-linear partitioning. In all experiments concentrations are increased at the prescribed rates until they reach four times their pre-industrial values.

### 2.3.3 Uncertainty sampling with FAIR

Uncertainty in the thermal response to radiative forcing typically tends to be the dominant factor in the uncertainty in the response of the global climate system to $CO_2$ emissions (Gillett et al., 2013). ECS and TCR co-vary in global climate models (Knutti et al., 2005; Millar et al., 2015), with TCR typically considered the more policy-relevant parameter and is better constrained by climate observations to date (Frame et al., 2006). Hence varying ECS alone in a probabilistic assessment risks introducing an implicit distribution for TCR that is inconsistent with available observations. Millar et al. (2015) observed that, within the coupled models of the CMIP5 ensemble, the TCR and the ratio TCR:ECS (referred to as the Realised Warming Fraction or RWF) are approximately independent. IPCC-AR5 provided formally assessed uncertainty ranges for TCR and ECS (Collins et al., 2013) but not for their ratio. RWFs for the CMIP5 models lie within the range 0.45-0.7, while observationally-constrained estimates typically lie in the upper half of this range (Millar et al., 2015).

We assess the impact of uncertainty in the FAIR parameters on the response to a 100GtC pulse emission of $CO_2$ in 2020 (against a background RCP2.6 concentrations (van Vuuren et al., 2011)) via a large ensemble (300 members) of draws from distributions representative of assessed uncertainty in these parameters. As IPCC-AR5 likely (>66% probability) ranges for a physical climate parameter attempt to capture structural uncertainties that might be present in all studies, IPCC-AR5 likely intervals are generally comparable to the 90% confidence intervals in the underlying studies as opposed to the central 66% of the distribution. IPCC-AR5 gives no assessment of the shape of the distribution associated with structural uncertainty as, by definition, this encompasses "unknown unknowns" that are not included in any model or study available. For quantitative modelling purposes, likely ranges are best interpreted as 5-95 percentiles of input distributions for IPCC-AR5 assessed parameters, provided a similar "structural degradation" is applied to interpret the 5-95 percentiles of output quantities as corresponding only to a likely range, propagating the possibility of structural uncertainty in the assessed parameter through the study.

We here assume a bounded (between 0 and 1) Gaussian distribution for RWF (with 5-95 percentiles of 0.45-0.75) and a log-normal distribution for TCR (with 5-95 percentiles of 1.0-2.5K), reproducing the positive skewness (fat high tail) of many estimated distributions for this parameter. A log-normal distribution has some theoretical justification as an appropriate shape for the distribution of a so-called "scale parameter" (one in which uncertainty increases with parameter size) which is arguably the case for TCR (Pueyo, 2012). Convolving these distributions gives a corresponding ECS 5-95 percentile range of 1.6-4.5K, in good agreement with the IPCC-AR5 assessed likely range (1.5-4.5K).

The short thermal response timescale, $d_2$, is an important determinant of the Initial Pulse-adjustment Time (IPT - the initial e-folding adjustment time of the temperature response to a pulse emission of $CO_2$ (NAS, 2016)). We sample $d_2$ using a log-normal distribution (as $d_2$ is a positive definite parameter) with 5-95% probability interval of 1.6-8.4 years, corresponding to the minimum of the CMIP5 range given in Geoffroy et al. (2013) as the $5^{th}$ percentile and the HadCM3 value of 8.4 years as the $95^{th}$ percentile. We consider uncertainties in the carbon cycle by sampling a single Gaussian random variable which is used to obtain draws from assumed Gaussian distributions of $r_0$, $r_T$ and $r_C$ for which the 5-95% probability intervals are equal

to +/- 13% of their default value (corresponding to a present-day iIRF$_{100}$ 5-95% probability interval of +/- 7 years). The 300 random draws from all of the above distributions are then used as model parameters for the integration of the 2020 100GtC pulse-response scenario described above, leading to a 300 member ensemble of climate outcomes indicative of the propagated uncertainty in the FAIR input parameters.

## 3    Results and discussion

### 3.1    The necessity for a state-dependent impulse-response model

When the AR5-IR model is integrated under estimated historical emissions from the Global Carbon Project (GCP) (Le Quéré et al., 2015) starting from an assumed quasi-equilibrium in 1850, atmospheric $CO_2$ concentrations increase faster than observed (figure 1a). This is indicative of the under-efficiency of the AR5-IR carbon sinks when continuously integrated over the observed period, resulting in a bias of over 30ppm in 2011 $CO_2$ concentrations. Similarly this under-efficiency of carbon sinks requires lower than observed emissions to simulate the observed timeseries of atmospheric $CO_2$ concentrations (figure 1b). Whilst sinks are maintained at their pre-industrial efficiency throughout (by definition) for the PI-IR model, it is only after approximately 1980 when the FAIR airborne fraction rises above the PI-IR airborne fraction (figure 1c). This arises due to a combination of a lower pre-industrial iIRF$_{100}$ in the FAIR model for a 100GtC pulse (see section 3.2) as $\alpha < 1$ in the FAIR pre-industrial state, annual emissions much less than 100GtC (reducing the iIRF$_{100}$ relative to a 100GtC pulse in FAIR but not PI-IR), and the PI-IR model (and the AR5-IR model) not capturing temporary reductions in the airborne fraction associated with volcanic-forced cooling (figure 1d) mediated through the temperature-induced feedback on the carbon-cycle in FAIR. Figure 1c shows large amplitude variations in the observed instantaneous airborne fraction that are likely to be driven in large part by unforced variability in the Earth-system and as such we would not expect these oscillations to be reproduced by any of the simple climate-carbon-cycle models. More complex carbon-cycle models are required to understand the drivers of these variations and any implications that they have for future carbon-cycle responses. Observed anomalies of global-mean temperature are reproduced well in the FAIR and PI-IR models (figure 1d), but present day warming is too large in the AR5-IR model, driven by substantially higher-than-observed present-day $CO_2$ concentrations.

Another key test of simple coupled climate-carbon-cycle models is the ability to replicate the response of ESMs to possible scenarios of future emissions. Commonly-used future scenarios are generally defined in terms of concentration pathways (Van Vuuren et al., 2011) and therefore do not have a model-independent set of emissions associated with them. In this paper we drive all three simple impulse-response climate-carbon-cycle models by a single set of emissions for each future scenario that are diagnosed from the MAGICC model (Meinshausen et al., 2011) in order to allow a comparison of simulated concentrations between simple models driven by identical inputs. MAGICC has been shown to be a good emulator of the CMIP5 ensemble and therefore offers a comparison by proxy to the projections of CMIP5 ESMs (Meinshausen et al., 2011). Whilst the PI-IR model might do a better job than the AR5-IR model of reproducing historical concentrations under high future emissions scenarios such as RCP8.5, it underestimates end of century concentrations, relative to MAGICC, to an even greater extent than the AR5-IR model (figure 2a) and concentrations fall from their peak even quicker than MAGICC under the high mitigation RCP2.6

scenario (figure 2b). The lack of saturation of carbon sinks in the AR5-IR model prevents the simulated concentrations keeping pace with MAGICC by the end of the $21^{st}$ century (under RCP8.5) despite having higher concentrations than MAGICC over the historical period and until approximately 2070. AR5-IR concentrations peak significantly higher than MAGICC under RCP2.6 and also decline faster after the concentration peak than simulated in MAGICC and FAIR. The deviation of both of these two models from MAGICC clearly indicates that any state-insensitive impulse-response model is therefore unsuitable, unless modified, for long integrations with historical and projected emissions.

The FAIR model compares well to MAGICC in both RCP8.5 and RCP2.6, scenarios that span the range of plausible future emissions trajectories. Concentrations simulated by FAIR are marginally higher than MAGICC after 2100 in RCP8.5, but the behaviour of MAGICC (or indeed any other model) under these more extreme forcing scenarios has not been verified. Additionally, concentrations in FAIR peak at a slightly lower value than MAGICC in the RCP2.6 scenario. Whilst comparing the performance of one simple model to another is not as rigorous a test of model performance as comparing directly to the behaviour of ESMs, it is encouraging that the FAIR model shows a close correspondence with a well-known and well-used simple model that has been used extensively to emulate the response of ESMs (Rogelj et al., 2012).

### 3.2 Response to pulse emission experiments

Figure 3 shows the response to a pulse-emission of $CO_2$ of differing magnitudes and against different backgrounds. For a 100GtC pulse of carbon set against an approximately present-day (389ppm) concentration background (PD100 - figure 3a), the FAIR simulated concentration anomaly associated with the pulse decays to 0.43 of its initial value after 100 years, slightly greater than the multi-model average of the ESM responses (0.41). The FAIR $iIRF_{100}$ of 50.5 years lies within the ESM multi-model spread (see table 2). Excluding temperature feedbacks on the carbon-cycle in FAIR (the "biogeochemically-coupled" version of FAIR - setting $r_T = 0$) increases the decay of the concentration response to the pulse over the century following the pulse emission, reducing the $iIRF_{100}$ by 10%. The AR5-IR, PI-IR and the FAIR model all show temperature anomalies due to the pulse initially adjusting rapidly followed by near-constant temperature over the remainder of the century as displayed by the Joos et al. (2013) multi-model mean (with internal variability superimposed on-top of this signal). Peak temperature anomaly is achieved after 12 years, consistent with the value of 10 years found by Ricke and Caldeira (2014).

Figure 3b and 3c show the response to a 100GtC and a 5000GtC pulse respectively (named PI100 and PI5000), applied in pre-industrial conditions. In figure 3c the pulse size is 50 times that in figure 3d, thus we divide the temperature response in the PI5000 experiment by 50 in figure 3c to allow the response per 100GtC to be visually comparable across the PD100, PI100 and PI5000 experiments. The 100GtC pre-industrial pulse decays faster than the present-day case (as in the ESMs and EMICs), due to reduced saturation of the land and ocean carbon sinks in the background state. FAIR simulates an $iIRF_{100}$ approximately 32% lower in the pre-industrial case (34.3 years) relative to the present day, lying just within the ensemble spread of PI100 $iIRF_{100}$ simulated by the models of Joos et al. (2013) (34-47 years). The magnitude of the FAIR simulated temperature response is similar in both the PD100 and PI100 cases due to the increased radiative efficiency of a pulse of $CO_2$ at lower background concentrations counteracting the faster decay of carbon out of the atmosphere. FAIR simulates a 100% increase of $iIRF_{100}$ in the 5000GtC pre-industrial pulse relative to the 100GtC pre-industrial pulse, consistent with the approximate doubling observed

in the ESMs. This clearly demonstrates that FAIR can capture the dependence of the pulse-response on pulse size as well as background conditions, whilst the AR5-IR model displays identical pulse response independent of pulse size or background conditions. The very rapid drawdown of $CO_2$ simulated by FAIR in the initial timestep following the 5000GtC pulse is in-part a function of the annual timestep in the model (carbon sinks remain at their pre-industrial efficiencies over the entirety of the first year despite accumulating a substantial amount of carbon over that period) and could be alleviated by using a smaller timestep.

Restricting temperature-induced feedbacks on the carbon-cycle does not result in a substantial reduction in the $iIRF_{100}$ for the pre-industrial 100GtC pulse experiment (the "fully-coupled" and "biogeochemically-coupled" experiments lie on top of each other in figure 3b), whereas a 13% reduction in $iIRF_{100}$ is observed for the Bern3D-LPJ model examined in Joos et al. (2013). This is due to the FAIR formalism, global mean temperature increase and cumulative carbon uptake increases $iIRF_{100}$ from a fixed pre-industrial value ($r_0$), therefore preventing substantial fractional differences between the "fully-coupled" and "biogeochemically-coupled" PI100 experiments as $iIRF_{100}$ remains close to $r_0$ throughout both these experiments. For the 5000GtC pre-industrial pulse experiment we see a reduction in the $iIRF_{100}$ (17%) associated with suppression of the temperature-induced feedbacks on the carbon cycle in FAIR, consistent with the approximate 15% reduction in $iIRF_{100}$ for Bern3D-LPJ. As the impact of the temperature-induced carbon-cycle feedbacks have only been assessed in a single EMIC, it is unclear how consistent or not FAIR is with the range of behaviour that would be simulated by the full ESM and EMIC ensemble.

In figure 4 we show emulations of the individual models in the Joos et al. (2013) ensemble using a single set of parameters for both the PD100 (figure 4a) and PI100 (figure 4b) simulations. In the fitting process we choose to fix the ratio of the $r_T$ and $r_C$ parameters to the default value (see section 2.3.1) as the fully coupled PD100 and PI100 experiments do not distinguish between temperature-induced and $CO_2$-induced feedbacks. Whilst significant diversity is seen in the range of responses to the PD100 and PI100 experiments across different ESMs/EMICs, attributable to a range of differences in carbon-cycle process representations within these models, variations in just a sub-set of the FAIR parameters are sufficient to span the ranges of responses in both the PD100 and PI100 experiments (fits are worse for models in which inter-annual variability is simulated), as well achieving the correct ratio between the PD100 and PI100 responses for models across the ESM/EMIC ensemble.

### 3.2.1 Temporal dependence of $CO_2$-induced warming on pulse size

Ricke and Caldeira (2014) used a version of the AR5-IR model to find that the maximum warming from a pulse emissions of $CO_2$ occurs approximately a decade after emission, but didn't account for important sensitivities of the timing of peak warming on the size of the emission pulse (Zickfeld and Herrington, 2015). Figure 5 shows that under a variety of magnitudes of present-day pulse emission magnitudes (from Herrington and Zickfeld (2014) - see section 2.3.1), the AR5-IR model warms rapidly to a near-term peak followed by a decline over the next century and then subsequently rises again as the long-timescale thermal response to radiative forcing begins to dominate the cooling effect of atmospheric $CO_2$ decay. The timing of the near-term warming peak is largely insensitive to the magnitude of the pulse emission in the absence of explicit feedbacks between the carbon-cycle and thermal climate system. Unlike in the AR5-IR model, FAIR shows a dependence of the temporal evolution

| Experiment - iIRF$_{100}$ (yr) | AR5-IR | PI-IR | FAIR | Joos et al. (2013) |
|---|---|---|---|---|
| PD100 | 52.5 | 40.8 | 50.5 | 52.4 ($\pm$11.3) |
| PD100 (no climate-carbon-cycle feedbacks) | - | - | 45.2 | - |
| PI100 | 52.5 | 40.8 | 34.3 | 40.6 ($\pm$4.2) |
| PI100 (no climate-carbon-cycle feedbacks) | - | - | 33.8 | 35.3* |
| PI5000 | 52.5 | 40.8 | 68.6 | 76.9 ($\pm$6.2) |
| PI5000 (no climate-carbon-cycle feedbacks) | - | - | 60.0 | 65.4* |

**Table 2.** iIRF$_{100}$ values in the experiments and models shown in figure 3. The star indicates that the values for suppressed climate-carbon-cycle feedbacks are calculated assuming the 13% (PI100) and 15% (PI5000) reductions in iIRF$_{100}$ from the fully coupled experiment observed from the Bern3D-LPJ model in Joos et al. (2013), the only model which conducted the experiments. Brackets indicate the +/- 2 standard deviation ranges around the multi-model mean in Joos et al. (2013).

of warming after present-day carbon pulses more similar to that seen in the UVic Earth System Climate Model (UVic-ESCM - see Eby et al. (2013)), in which warming reaches a maximum later for larger pulse sizes (black lines in figure 5), as the balance between carbon-cycle cooling and long-timescale thermal warming takes centuries to reach equilibrium. FAIR does simulate a small near-term warming peak for the smallest pulse size, although warming only decreases by less than 2% before

subsequently rising again. As the magnitude of future cumulative emissions are uncertain and could exceed multiple TtC under minimal climate policy scenarios (e.g. RCP8.5) correctly capturing the dependence of peak warming on injection size in simple climate-carbon-cycle models is important for correctly assessing the multi-millennial impacts of climate policy (Clark et al., 2016).

### 3.3   Response to idealised concentration increase experiments

The set of "fully-coupled", "biogeochemically-coupled" and "radiatively-coupled" experiments from Gregory et al. (2009) (see section 2.3.2) can help to isolate the contributions from temperature-induced and direct carbon-induced effects on overall carbon-cycle feedbacks. Coupling between temperature changes and the carbon-cycle in the FAIR model acts to suppress carbon uptake, shown by the difference between the "fully-coupled" and "biogeochemically-coupled" (thick and thin blue lines in figure 6a) under a 1%/yr $CO_2$ concentration increase scenario, consistent with the behaviour shown by the ESMs from

Arora et al. (2013) (thick and thin pastel coloured lines). This is a mechanism that is absent (by construction) in the AR5-IR model. Figure 6b shows $C_{\text{acc}}$ as a function of atmospheric $CO_2$ concentration for several rates of prescribed concentrations increase: again, the FAIR model captures the concave-downward form of this diagnostic shown by the ESM ensemble for all rates of concentration increase, in contrast to the AR5-IR model.

Oceanic carbon-cycle feedbacks are almost exclusively driven by biogeochemical effects (Glotter et al., 2014). However,

simple global climate-carbon-cycle models need to also capture dependencies of the land carbon uptake on warming. Aside from 3 ESMs that display global-mean carbon-cycles relatively insensitive to warming, figure 6c shows a relationship between temperature increases and the size of the carbon outgassing back to the atmosphere in the 'radiatively-coupled' FAIR experi-

ment similar, in both shape and magnitude, to that displayed in the ESMs under the 1%/yr concentration increase experiment of Arora et al. (2013). $1\%\mathrm{yr}^{-1}$, $0.5\%\mathrm{yr}^{-1}$ and $2\%\mathrm{yr}^{-1}$ experiments in FAIR all lie along the same line in figure 6c, indicating minimal scenario dependence of this effect in FAIR, in contrast to the two ESMs analysed in Gregory et al. (2009).

As simulated by FAIR, the initial decrease in cumulative airborne fraction (the fraction of all past emissions remaining in the atmosphere) followed by a subsequent increase is a feature of the response of many ESMs under a 1%/yr increasing $CO_2$ scenario (figure 6d). In contrast, the IPCC-AR5 model shows a steady decrease in the cumulative airborne fraction with higher concentrations due to the state-invariant rates at which a pulse of carbon is removed from the atmosphere. The initial decrease in cumulative airborne fraction followed by subsequent increase can be understood in terms of the saturation of carbon sinks. If atmospheric anomalies of carbon decay with fixed timescales, $\tau_i$ (as in the AR5-IR model case), then the instantaneous airborne fraction remains constant in time, which necessarily means that cumulative airborne fraction must decline over time (as emissions from previous years decay further, so the cumulative fraction of the emitted carbon continually decays from the instantaneous airborne fraction). However, if carbon sinks become saturated, the instantaneous airborne fraction would be expected to increase with time (this is represented in the FAIR model by increases to the decay timescales through the parameterised increase in $\mathrm{iIRF}_{100}$). Therefore, as more recent emissions (which increase monotonically with time in the prescribed concentration increase experiments) have a higher instantaneous airborne fraction, the initial decrease in cumulative airborne fraction slows, stops and then this fraction subsequently begins to increase as the accelerating saturation becomes the dominant effect.

## 3.4 Probabilistic parameter sampling within the FAIR model

The impulse-response formulation of the physical climate response to radiative forcing used by both the AR5-IR and FAIR models (equation 3) offers a convenient structure for simply sampling plausible ranges of TCR and ECS. Figures 7a and 7b show the isolated impact of thermal response uncertainty under idealised prescribed concentration scenarios, namely a 1%/yr $CO_2$ concentration increase (figure 7a) and an instantaneous quadrouping of $CO_2$ concentrations from pre-industrial values (figure 7b). A unique combination of TCR and ECS (for fixed response time-scales $d_j$) are associated with a unique combination of the model parameters $q_j$ (equations 4 and 5). The blue shading in panels a) and b) of figure 7 show the simulated temperature response for the likely range of TCR and ECS as assessed by IPCC-AR5 (TCR: 1.0-2.5K and ECS: 1.5-4.5K). Expressed in terms of $q_j$, the top end of the IPCC-AR5 thermal uncertainty range (TCR=2.5K, ECS=4.5K) corresponds to $q_1 = 0.57\mathrm{KW}^{-1}\mathrm{m}^2$ and $q_2 = 0.63\mathrm{KW}^{-1}\mathrm{m}^2$, with the lower end (TCR=1.0K, ECS=1.5K) corresponding to $q_1 = 0.14\mathrm{KW}^{-1}\mathrm{m}^2$ and $q_2 = 0.26\mathrm{KW}^{-1}\mathrm{m}^2$. The thermal response model given by equation 3 fully spans the range of responses seen in the CMIP5 ensemble for these fixed concentration integrations under with the ranges of parametric uncertainty given above. AR5-IR results are not shown in figure 7a and 7b as under prescribed $CO_2$ concentrations uncertainty in warming arises from thermal response uncertainty only - which is identical in the FAIR and AR5-IR/PI-IR models.

A robust feature of the carbon-cycle response in all ESMs is an increase in the cumulative airborne fraction over time associated with a saturation of carbon sinks (upward curving black lines in figure 7c). Unlike the AR5-IR model, which displays a slowly declining cumulative airborne fraction over time due to the state-independence of its response function,

correlated and time invariant perturbations of +/-13% to the $r_0$, $r_T$ and $r_C$ parameters (combined with perturbations to $q_1$ and $q_2$ consistent with the IPCC-AR5 likely ranges) in the FAIR model approximately spans the range of responses seen in the CMIP5 models under a $1\%\text{yr}^{-1}$ concentration increase scenario (blue shading in figure 7c) and displays increasing cumulative airborne fraction over time at both ends of the uncertainty range.

Crucially, the FAIR model also captures the approximately linear relationship (to first order) between cumulative carbon emissions and human-induced warming (figure 7d) that was highlighted in the IPCC 5th Assessment, and is becoming an integral part of climate change policy analysis (Millar et al., 2016). When integrated with parameter settings given in section 2, FAIR has a Transient Response to Cumulative Emissions (TCRE[4]) =1.3K/TtC (thick blue line in figure 7d). As there is some downward curvature apparent in the FAIR relationship between warming and cumulative emissions across the range of

cumulative emissions shown in figure 7d (as also displayed by the CMIP5 ESMs) the numerical value of the TCRE is only exactly valid for the first 1000GtC of an emissions injection. Perturbations to the model parameters as described above (and identical to Figure 7c) allow the IPCC-AR5 likely TCRE range of 0.8-2.5K/TtC to be spanned (Figure 7d). In contrast, the AR5-IR model, with a constant airborne fraction, shows a more-pronounced concave-downward shape in the plot of realised warming against cumulative carbon emissions, as the decline of the cumulative airborne fraction is unable to compensate (as

it does in more complex models) for the logarithmic relationship between $CO_2$ concentration and radiative forcing (Millar et al., 2016). The curvature in the relationship between warming and cumulative emissions in FAIR displays most prominent curvature at high cumulative emissions, consistent with the behaviour of ESMs (Leduc et al., 2015).

Figure 8 shows the combined effect of uncertainty in the FAIR model parameters, both thermal parameters (the joint distribution of TCR and ECS is shown in figure 8a) and carbon-cycle feedback parameters. Representative uncertainty in FAIR

parameters (see section 2.3.3) propagates into an emergent 5-95% range (based on 300 draws from the input parameter distributions) for TCRE (figure 8b) of 1.0-2.5K/TtC when integrated under a 1%/yr $CO_2$ concentration increase scenario (figure 8b), broadly consistent with the IPCC-AR5 *likely* range (0.8-2.5K/TtC). The temperature anomaly associated with a 100GtC pulse in 2020 against an RCP2.6 background is shown in figure 8c. Across the ensemble a range of responses in both magnitude and shape are observed. However, we consistently observe a rapid warming on the order of a decade followed by an approximate

warming plateau (at differing values) that persists for a century or more.

## 4  Conclusions

In this paper we have presented a simple Finite Amplitude Impulse Response (FAIR) climate-carbon-cycle model, which adjusts the carbon-cycle impulse-response function based on feedbacks from the warming of the climate and cumulative $CO_2$ uptake through a parameterisation of the 100-year integrated impulse-response function, $\text{iIRF}_{100}$. We use this metric of carbon-

cycle response as a parallel to those used to assess the thermal response to radiative forcing, namely the Transient Climate Response (TCR) and the Equilibrium Climate Sensitivity (ECS).

---

[4]TCRE is defined as the annual mean global surface temperature change per unit of cumulative $CO_2$ emissions, usually 1000GtC, in a scenario with continuing emissions (Collins et al., 2013).

We have shown that including both explicit $CO_2$ uptake- and temperature- induced feedbacks are essential to capture ESM behaviour. Neglecting temperature-induced feedbacks on the carbon-cycle would prevent a simple climate-carbon-cycle model being able to capture the important dependences of carbon uptake on warming displayed in 'radiatively-coupled' ESM experiments, largely driven by responses of the land carbon-cycle. Similarly, neglecting $CO_2$ uptake-feedbacks would fail to incorporate well-understood physical mechanisms governing the response of ocean carbonate chemistry to anthropogenic $CO_2$ emissions.

Important dependences of the carbon-cycle response to pulse size, background conditions and the suppression of temperature-induced feedbacks are generally well captured by the FAIR model. The inclusion of climate-carbon-cycle feedbacks in the FAIR model offers an improvement on several simple and transparent climate-carbon-cycle models that have been proposed for policy analysis which either incorporate no feedbacks on the carbon-cycle or do not fully capture the operation of these feedbacks in ESMs. Successfully emulating the approximate balance between warming-induced and biogeochemically-induced contributions to carbon-cycle feedbacks could be important for integrated assessment of solar radiation management scenarios and mitigation scenarios in which the balance of contributions to warming from $CO_2$ and non-$CO_2$ sources changes significantly in the future.

Throughout this paper we have used iIRF$_{100}$ as a central metric of the climate system response. This represents an inherent value choice about the timescales of the coupled climate-carbon-cycle system prioritised for correct representation in a simple climate-carbon-cycle model. A time horizon of 100 years captures important aspects of the climate response to a pulse emission of $CO_2$ relevant over typical economic discounting timescales, whereas a longer time horizon could be used to prioritise the millennial timescale response. For studies that are primarily focused on the climate response over multi-millennial timescales it may be most appropriate to retune the values of the FAIR model to capture the dependencies shown by ESMs/EMICs over this time period (e.g. Zickfeld et al. (2013)). However, over these time periods Earth system feedbacks not simulated in ESMs and EMICs may become important, questioning the validity of emulating particular aspects of very long-term ESM behaviour. Although a useful composite metric for the coupled climate-carbon-cycle system already exists, the Transient Climate Response to Cumulative Emissions (TCRE), future studies of carbon cycle behaviour could usefully report on ranges of iIRF$_{100}$, and importantly for carbon cycle feedbacks, the evolution of this metric over time under specific emissions scenarios, in order to isolate the changing response of the carbon cycle and to enable emulators such as FAIR to span the ranges and capture the dependencies of iIRF$_{100}$ that are observed in state-of-the-art models.

We believe that the FAIR model could be a useful tool for offering a simple and transparent framework for assessing the implications of $CO_2$ emissions for climate policy analyses. It offers a structure that both replicates the essential physical mechanisms of the climate system's response to cumulative emissions, whilst at the same time it can easily be modified to sample representative climate response uncertainty in either the thermal climate response component, the unperturbed carbon-cycle or the coupled climate-carbon-cycle response to anthropogenic $CO_2$ emissions. Tuning of parameters within FAIR allows the range of ESM behaviour to be emulated whilst maintaining the physically-understood dependency of the carbon-cycle pulse-response on background conditions and pulse size exhibited by a particular ESM. This model structure could thus be adapted to be an effective emulator of CMIP6 ESM responses under a variety of scenarios.

*Author contributions.* RJM, ZRN and MRA developed the FAIR model formulation. PF and MRA identified the need for the feedback term in the AR5-IR model while RJM developed the final formulation. MRA designed the tests and RJM made the figures, except Figure 4 which was made by ZRN. RJM wrote the first draft of the manuscript and all authors contributed to the editing and revisions of the paper.

## 5  Data Availability

5  A repository containing the code for the FAIR model can be found at: https://github.com/OMS-NetZero/FAIR

*Acknowledgements.* We would like to thank Victor Brovkin, Elisabeth Moyer and several other anonymous reviewers for their useful comments on our manuscript. RJM and MRA would like to acknowledge financial support from the Oxford Martin School and RJM and PF from the Natural Environment Research Council project NE/P014844/1.

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

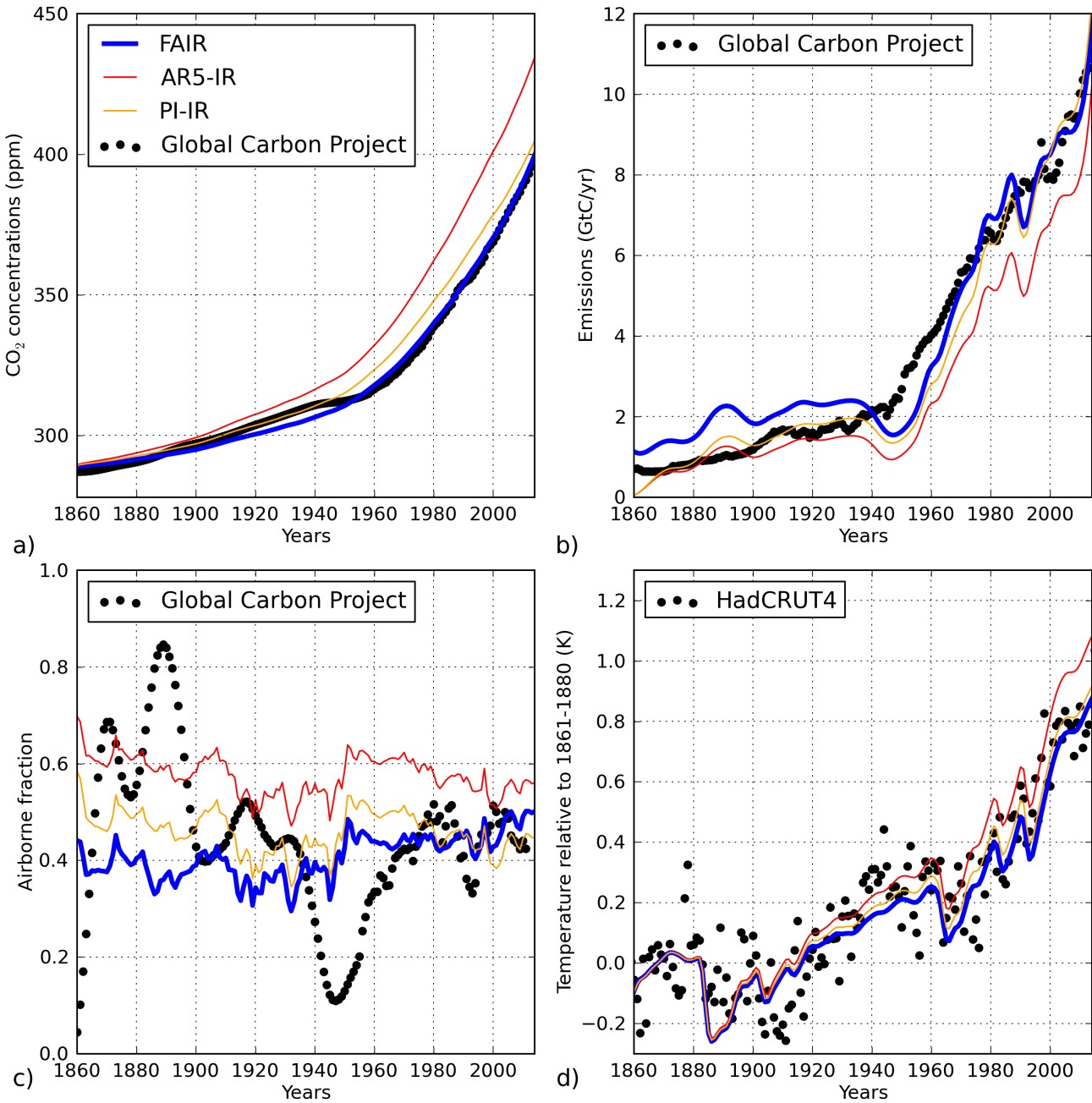

**Figure 1.** Historical validation of the FAIR (blue), AR5-IR (red) and PI-IR (orange) models. Panel a) shows simulated $CO_2$ concentrations when driven by historical emissions (and historical non-$CO_2$ forcing) as estimated by MAGICC. Panel b) shows diagnosed $CO_2$ emissions consistent with historical concentrations. Panel c) shows the evolution of annual airborne fraction (smoothed with a 7-year running mean for the observations), and d) the warming anomaly when driven by historical emissions. Historical observations are shown as black dots in all panels. Panels a), b) and c) show Le Quéré et al. (2015) and HadCRUT4 (Morice et al., 2012) is shown in d). All simulations are commenced from assumed quasi-equilibrium carbon-cycle states in 1850.

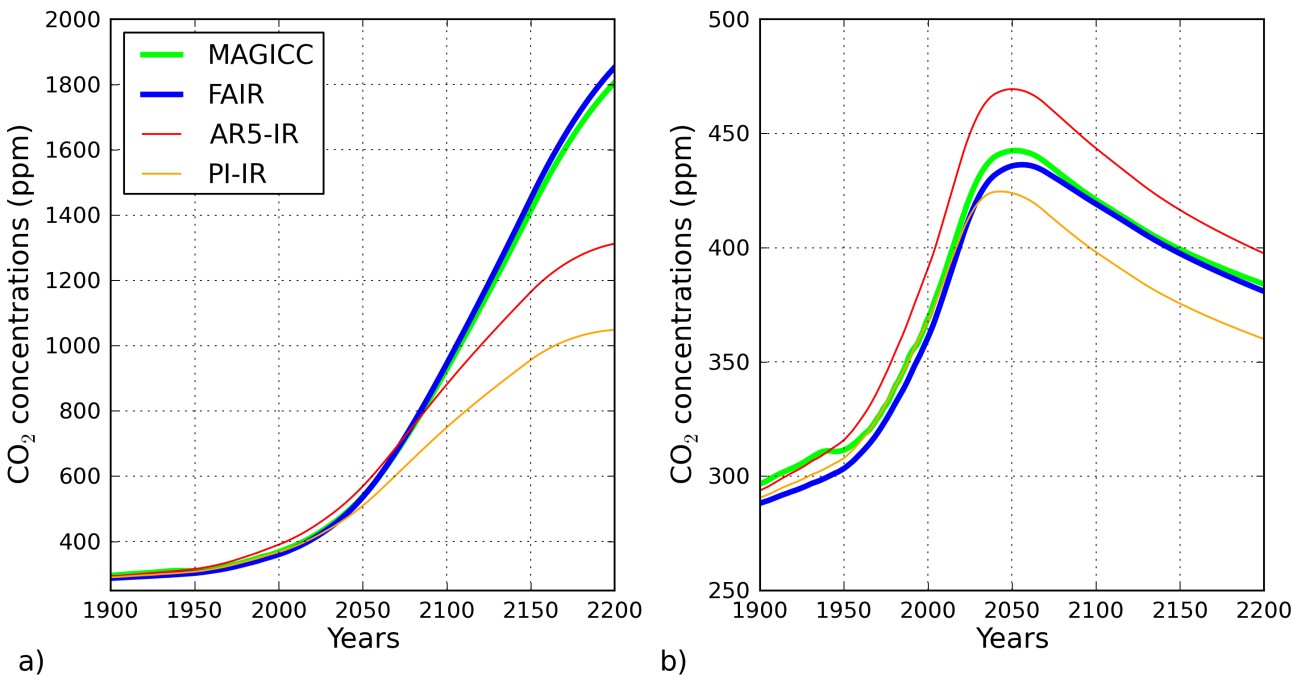

**Figure 2.** $CO_2$ concentrations under RCP8.5 (a) and RCP2.6 (b). FAIR (blue), AR5-IR (red), PI-IR (orange) and MAGICC (green) are shown in both panels.

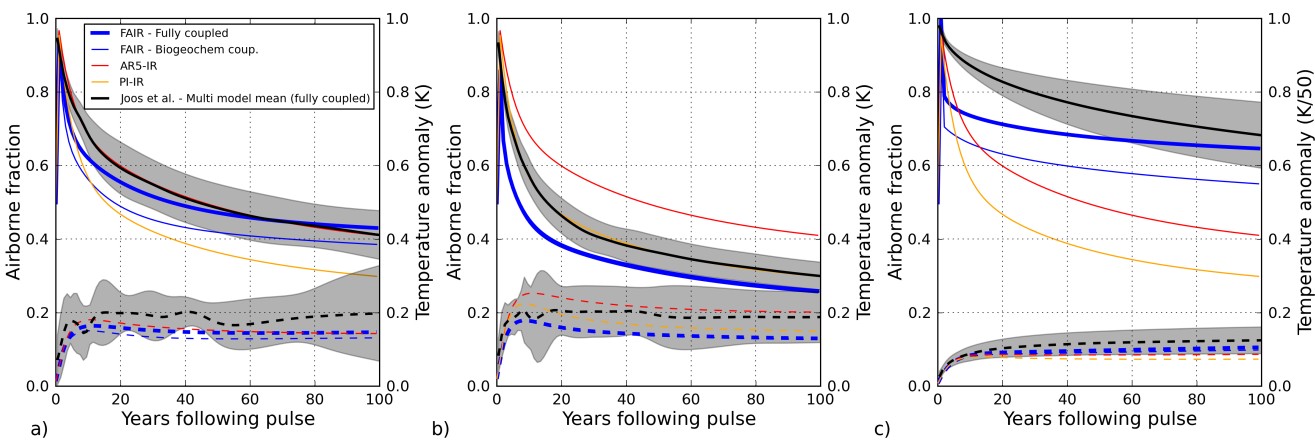

**Figure 3.** Response to pulse emission experiments of Joos et al. (2013). Panel a) shows the response to a 100GtC imposed on present-day (389ppm) background conditions (PD100 experiment), panel b) the response to a 100GtC pulse in pre-industrial conditions (PI100 experiment) and panel c) the response to a 5000GtC pulse in pre-industrial conditions (PI5000 experiment) with the warming normalised by the increase in pulse size between panels b) and c). Airborne fraction (left hand axis) is represented by solid lines in all panels and warming (right hand axis) by dashed lines. FAIR is shown as thick blue lines, AR5-IR as red and PI-IR as orange. The black lines in all panels shows the Joos et al. (2013) multi-model mean for airborne fraction (solid) and warming (dashed), with the grey shading indicating one standard deviation uncertainty across the ensemble. Thin blue lines denote the biogeochemically-coupled version of FAIR.

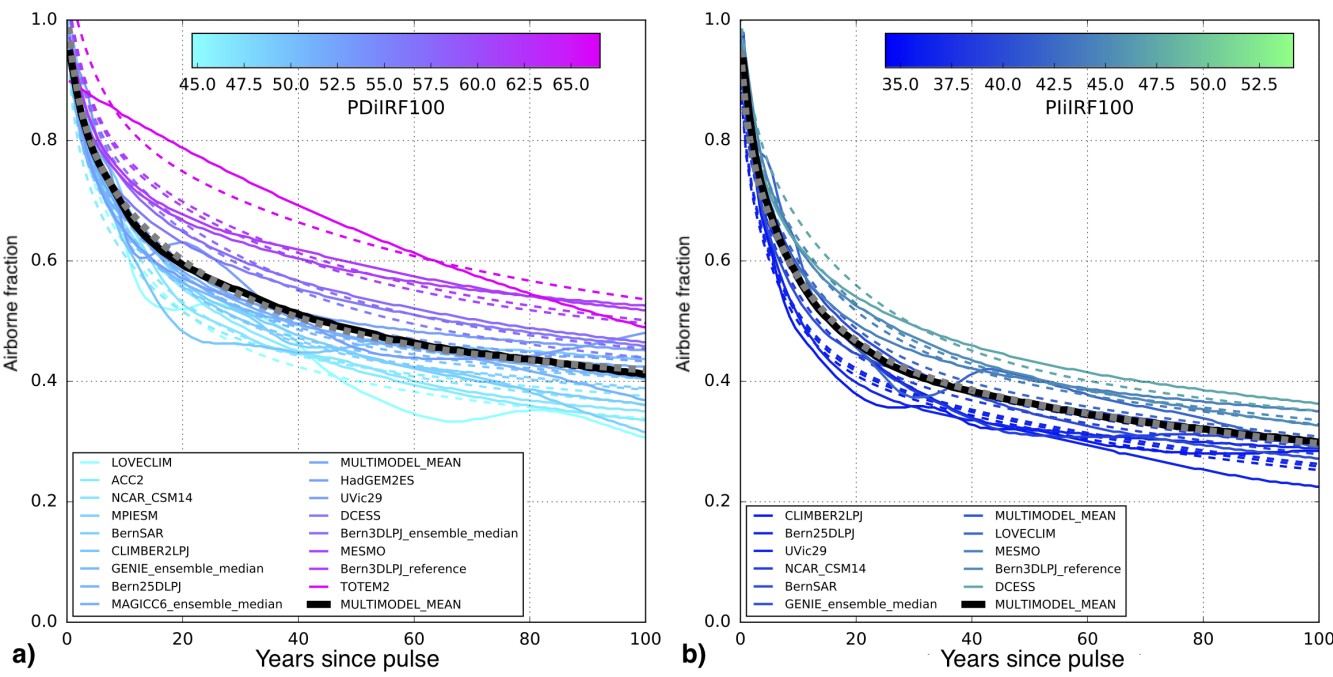

**Figure 4.** Fitting individual models from Joos et al. (2013) with FAIR. Panel a) shows the remaining airborne fraction for the PD100 experiment and panel b) for those models that additionally completed the PI100 experiment. Solid lines show the original model response coloured by the $iIRF_{100}$ values. Emulations with FAIR are shown by the same coloured dashed lines. The multi-model mean is shown by a solid black line with the FAIR fit denoted by a dashed grey line.

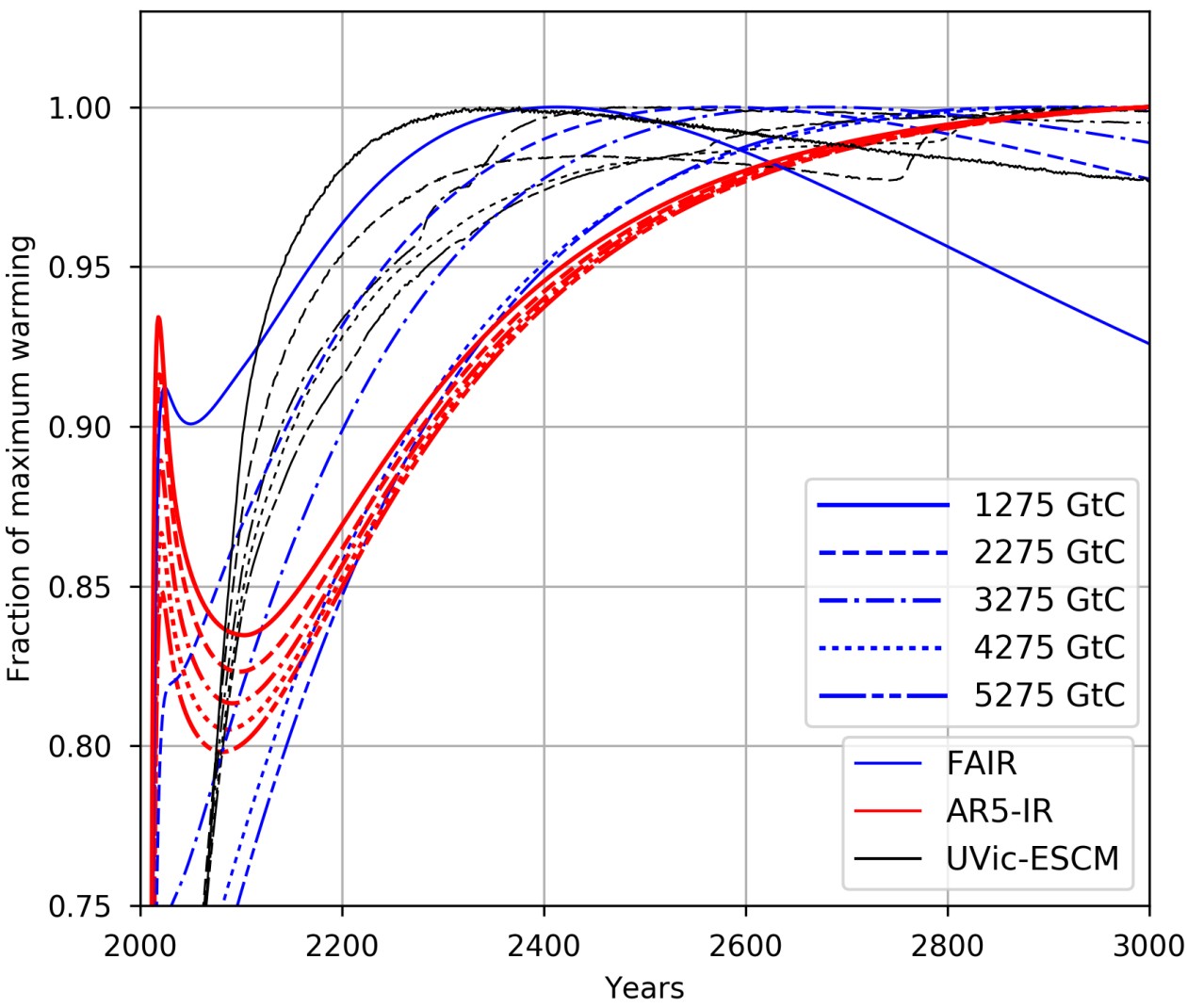

**Figure 5.** Dependency of the timing of maximum warming on pulse size. Global mean surface temperature response is expressed as a fraction of the maximum warming under the pulse experiments of Herrington and Zickfeld (2014). Responses are shown for the FAIR (blue), AR5-IR (red) and UVic-ESCM (black). Different cumulative emissions totals (see Herrington and Zickfeld (2014)) are denoted by different line styles.

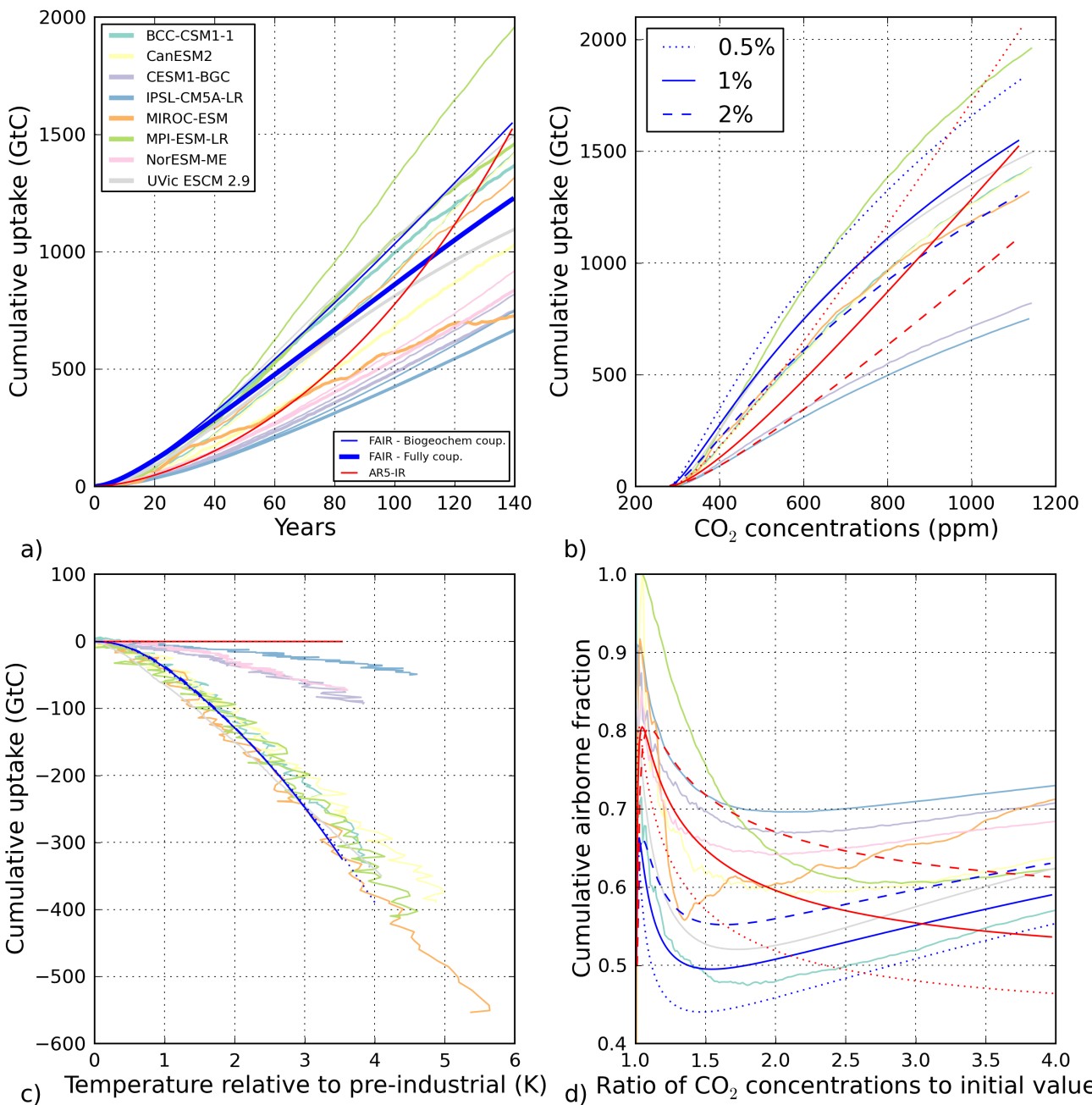

**Figure 6.** Response to idealised concentration increase experiments from Gregory et al. (2009) for the FAIR (blue) and AR5-IR (red) models. Light pastel colours show the ESMs from Joos et al. (2013) for the 1%/yr concentration increase scenario only. Panel a) shows the cumulative ocean and land carbon uptake over time in the "fully coupled" $1\%\mathrm{yr}^{-1}$ concentration increase scenario. Panel b) shows the evolution of cumulative ocean and land carbon uptake as a function of atmospheric concentration in the "biogeochemically coupled" experiment for $1\%\mathrm{yr}^{-1}$ (solid), $2\%\mathrm{yr}^{-1}$ (dashed) and $0.5\%\mathrm{yr}^{-1}$ (dotted) experiments. Panel c) shows the cumulative uptake as a function of temperature in the "radiatively coupled" experiments. Panel d) shows the evolution of the cumulative airborne fraction as a function of the proportional concentration increase for the "fully coupled" experiments.

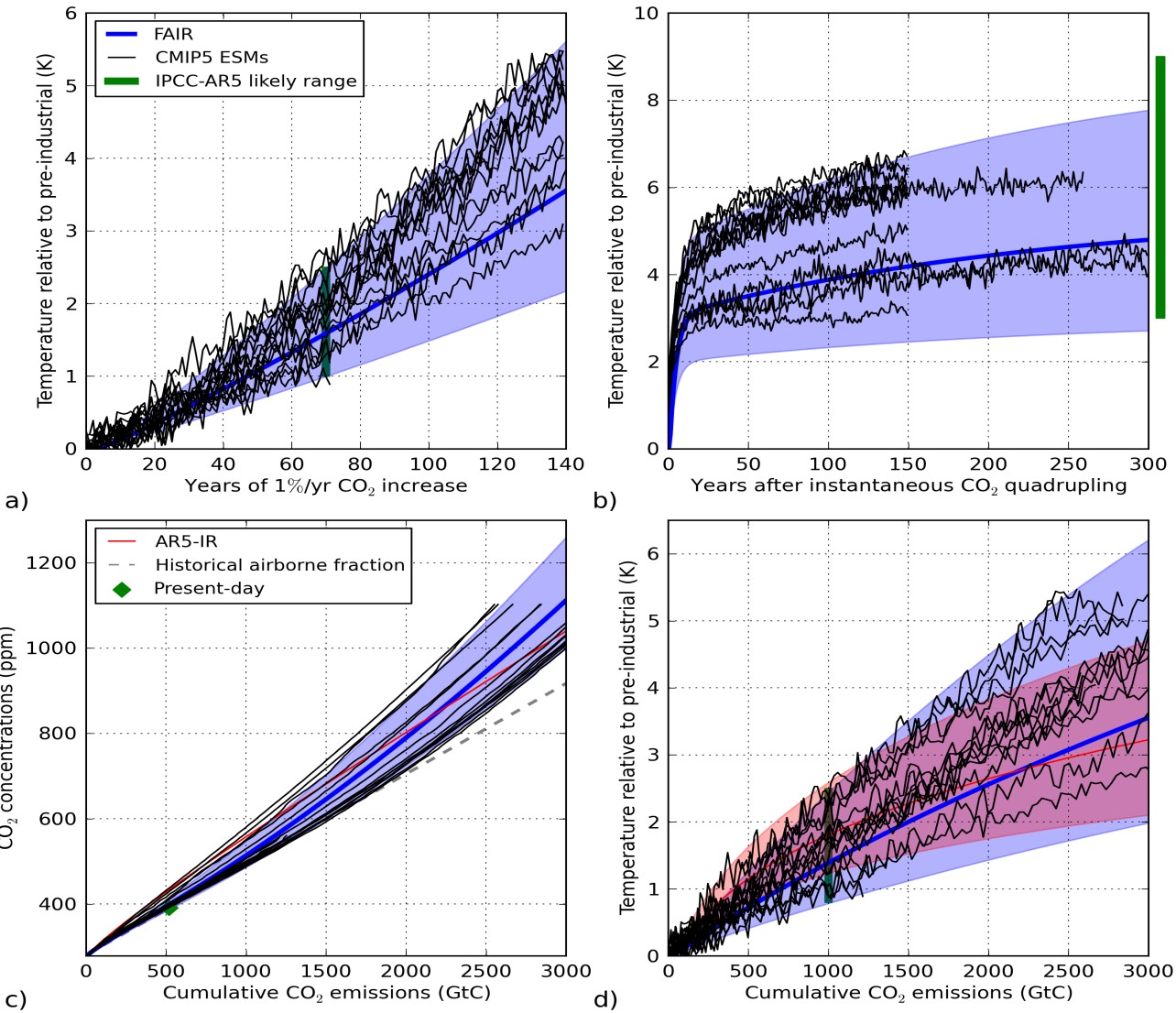

**Figure 7.** Climate response uncertainties in the FAIR (blue), AR5-IR (red) and CMIP5 (black) models. Panel a) shows the temperature responses to a 1%/yr concentration increase scenario. The green bar indicates the IPCC-AR5 TCR likely range. The blue shading in panels a) and b) shows the response of FAIR under IPCC-AR5 upper and lower likely TCR and ECS ranges. Panel b) shows the responses to an instantaneous quadrupling of atmospheric $CO_2$ which is held fixed subsequently. The green bar indicates the assessed equilibrium warming compatible with the IPCC-AR5 ECS likely range. Panel c) shows concentrations as a function of cumulative emissions in the 1%/yr scenario. The plumes in panels c) and d) show the FAIR simulated response under the IPCC-AR5 likely TCR and ECS ranges, with an additional +/- 13% perturbation to the $r_0$, $r_T$ and $r_C$ parameters for the high/low end response. The dashed grey line indicates a constant cumulative airborne fraction that is consistent with the present-day state of the climate system (green diamond). Panel d) shows warming as a function of cumulative emissions in the 1%/yr scenario. The green bar shows the IPCC-AR5 likely 0.8-2.5K/TtC assessed range for TCRE.

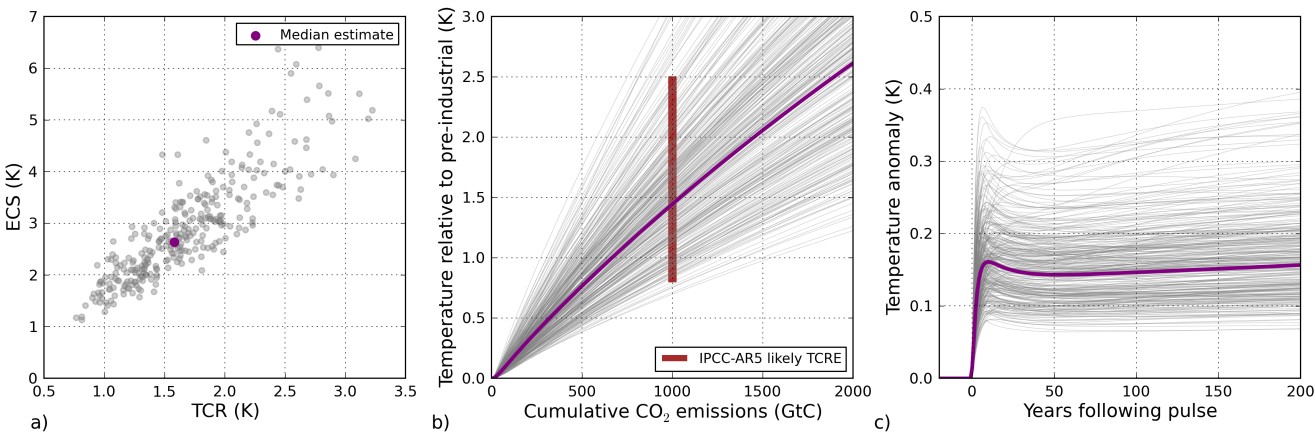

**Figure 8.** Probabilistic sampling in the FAIR model. Grey lines show 300 random draws from the input parameter distributions, as described in section 3.4. Panel (a) shows the joint distribution of TCR and ECS. Panel (b), warming as a function of cumulative emissions in the $1\%\mathrm{yr}^{-1}$ concentration increase experiment. The brown bar in panel b) represents the IPCC-AR5 likely TCRE range. Panel (c), the warming response to a 100GtC pulse emitted in 2020 on top of the MAGICC-derived RCP2.6 emissions. The purple dot (a) and lines (b and c) represent the median parameters of the distributions.