# Peer review of "A modified impulse-response representation of the global near-surface air temperature and atmospheric concentration response to carbon dioxide emissions"

_Atmospheric Chemistry and Physics, 2016_

## Referee Comment (RC1) · Anonymous Referee #1 · 5 Jul 2016

The paper presents a modification of the impulse response function model of Joos et al. (2013) used in the last IPCC report to convert $CO_2$ emissions into atmospheric $CO_2$ concentrations. The modifications consist of (i) making the "decay" time parameters $\tau_i$ dependent on model state, and (ii) changes in model parameters. After determining the new parameters, the new model (which the authors call "FAIR") is exposed to several tests (response to different sizes of emission pulses, comparison with historical atmospheric $CO_2$ development, comparison with results from two other carbon cycle models (BEAM, MAGICC)). In addition the authors demonstrate that by varying model parameters, ensembles of scenarios can be constructed with their model that account for the IPCC likely range of transient climate sensitivity.

[Figure]

The authors motivate the need for their modified impulse response function model by claiming "*This extension is necessary because the use of a state-insensitive impulse-response model cannot simultaneously reproduce the relationship between emissions, concentrations and temperatures seen over the historical period and the projected response over the 21$^{st}$ century to both high-emission and mitigation scenarios estimated from more complex models.*" (p. 2, lines 23ff). While it is true that the Joos et al. (2013) model is not accounting for a state dependence, it got not really clear to me from the paper whether this unability of 'simultaneous reproduction' is true or not. But anyway, since this claim is essential for justifying the study as a whole, this point had to be demonstrated right at the beginning of the paper. But such a demonstration is missing. Accordingly, for me the authors failed to make clear why I should read their study.

Moreover, the evaluation of the new model remains vague. Once parameters have been found for their new model (p. 4, lines 6-8) (how have those parameters been determined?), the results of several impulse experiments are discussed in section 3, resulting in questionable claims on the quality of their new model like "*consistent with corresponding ratio in the most detailed ESMs*" (p. 6, lines 23f) (what means 'most detailed'?) and "*the FAIR model can capture the dependence of the pulse-response on pulse size*" (p. 6, line 28) (what means 'capture'? In comparison to what?) or "*it is encouraging that the FAIR model shows a close correspondence with a well-known and well-used simple model [=MAGICC] that has been used extensively to emulate the response of ESMs*" (p. 8, lines 7f) ('encouraging' is nice but not convincing). Generally, a clear strategy for model evaluation is missing. In particular, it is not well specified which model simulations are meant to be emulated by FAIR – the specification in the abstract remains vague ("*several idealised experiments performed with more complex models*"). If I understand the intention of the study right, those models contributing to the carbon studies in CMIP5 are meant to be well emulated by the new FAIR model. But this model ensemble is not showing up in the plots (except a few graphs including results from BEAM and MAGICC). Data for comparison would have been available in international archives, and if also pulse experiments with these models would have

been necessary for comparison, Joos et al. could have been contacted for data from their 2013 study.

Generally, I think that the state dependence is an interesting addition to the original impulse response model, but the evaluation of the resulting new model is lacking rigor. I guess that by making the study more targeted and rigorous, the resulting paper would be very different. Considering all this, I suggest to reject the paper.

---

## Author Comment (AC1) · 13 Jul 2016

[12pt, letterpaper]article

Despite the reviewers recommendation we believe that the points raised can be relatively simply dealt with and would hope the making the changes outlines through the in-line responses to the specific points raised below would change the reviewers recommendation.

*The authors motivate the need for their modified impulse response function model by*

[Figure]

*claiming "This extension is necessary because the use of a state-insensitive impulse response model cannot simultaneously reproduce the relationship between emissions, concentrations and temperatures seen over the historical period and the projected response over the 21st century to both high-emission and mitigation scenarios estimated from more complex models." (p. 2, lines 23ff). While it is true that the Joos et al. (2013) model is not accounting for a state dependence, it got not really clear to me from the paper whether this unability of 'simultaneous reproduction' is true or not.*

Figure 4 shows that the AR5-IR model (red) fails to reproduce observed concentrations over the historical period when integrated with historical emissions (panel a) and fails to reproduce historical emissions when emissions consistent with historical concentrations are derived from the model (panel b). A substantial error of nearly 30 ppm is seen in figure 4a, representing a large difference from the correct climate state. We also show in figure 5 how the AR5-IR model fails to produce similar behavior to the MAGICC model under the RCP8.5 and RCP2.6 scenario. The MAGICC model is approximately consistent with the response of the ESMs in CMIP5. We agree with the reviewer that this point could be made more prominently and would emphasize this point more in the introduction and change the ordering of our results section in a revised version of the manuscript. We also propose to introduce a new figure to demonstrate that the problem with the AR5-IR model is structural: despite the number of tunable parameters in this model, no combination of parameters can simultaneously reproduce the historical period and projected changes over the 21st century without the introduction of a state dependence. We will post a draft version of this figure on this discussion as soon as possible.

*"it is encouraging that the FAIR model shows a close correspondence with a well-known*

*and well-used simple model [=MAGICC] that has been used extensively to emulate the
response of ESMs" (p. 8, lines 7f) ('encouraging' is nice but not convincing).*

We agree with the reviewer that agreement between two simple models of the climate system is not a sufficient evaluation of model skill, but given how extensively the MAGICC model is used, an ability to reproduce its behaviour might be argued to be a necessary condition for any simpler model. We will clarify this in revision.

*In particular, it is not well specified which model simulations are meant to be emulated
by FAIR.*

We intend for the FAIR model to reproduce features of the climate response to CO2 as shown in ESMs (e.g. p. 2, l. 19-20). We have often referred to "more complex/comprehensive models" in the text to indicate that we are attempting to emulate behavior simulated by both ESMs and EMICs (e.g. p. 2 l. 4) but we agree that this wording is ambiguous, particularly in the abstract, and could be easily rectified in a revised version of the manuscript.

We propose to add an additional figure demonstrating how the FAIR model can be used to emulate the results of more complex models in the Joos et al (2013) model intercomparison, but it must be emphasized that this represents only one of a large range of possible tuning exercises. No simple model or emulator should be assumed to be a substitute for more comprehensive models simply because it reproduces their behavior under a selected scenario or experiment: our objective here is to demonstrate that the FAIR model is potentially able to reproduce the results of more complex models, but for

any particular application, the model should be tuned to relevant simulations. Hence we would argue that providing "definitive" parameter values for the FAIR model would be misleading. We will post a draft version of this figure on this discussion as soon as possible.

*the results of several impulse experiments are discussed in section 3, resulting in questionable claims on the quality of their new model like "consistent with corresponding ratio in the most detailed ESMs" (p. 6, lines 23f) (what means 'most detailed'?)*

We intended consistency to refer to the quantative statement in the following clause, namely, "with its value of 36 years within the 34-47 years range of the ESMs". We agree with the reviewer that the word 'detailed' adds no extra information, introduces confusion and can be simply removed in a revised version of the manuscript. We agree with the author that adding ranges of the pulse responses in ESMs from Joos et al (2013) on figure 3 would aid the discussion of figure 3 in the text, and could be easily implemented.

*"the FAIR model can capture the dependence of the pulse-response on pulse size" (p. 6, line 28) (what means 'capture'? In comparison to what?)*

We will replace the word "capture" with the word "reproduce" if that is clearer. A note will be added to the text on p. 6 to indicate that Joos et al (2013) found the iIRF100 to decrease by 40% between the 5000GtC and 100GtC preindustrial baseline pulse (this

would be in addition to the ranges of response in Joos et al being added to figure 2 as discussed above).

---

## Referee Comment (RC2) · Anonymous Referee #2 · 17 Jul 2016

"A modified impuse-response representation of the global response to carbon dioxide emissions" by R.J. Millar et al., describes a simplified impulse-response model to simulate the airborne fraction of $CO_2$ and subsequent temperature response to $CO_2$ emissions. The goal of a simple model with which to examine probabilistic responses to $CO_2$ emissions is worthwhile, but the paper has major deficits that must be reconciled before the work is accepted for final publication. Given the current state of the paper, it is difficult to assess the scientific significance of this model.

First, the authors must provide a better literature review that contains brief descriptions of the other models to which their FAIR model is compared, including the IPCC simple model and the BEAM model, which is introduced rather abruptly in the results section.

The type of research that would be enabled by such a simple representation of climate-carbon couplings is likely to be interdisciplinary in nature, and therefore the tools that are introduced in this manuscript must be better explained for their paper to be of broad utility to the climate community. At the end of the introduction, the authors propose to "extend" on the IPCC-AR5 model, but I don't think they have fully introduced this model, or described why they think it is deficient.

Second, the authors must provide a better description of their own FAIR model. The time constants should be better described, the carbon reservoirs used in the model should be named, and all the variables used in equations 1-5 should be described and the appropriate units should be listed. Although a table would be helpful for this, the authors do not even describe all the variables in-line. The authors need to provide a brief explanation as to how the new values for ECS and TCR were arrived at, rather than assuming readers of this paper will consult the author's 2015 paper.

Third, the discussion of the figures, which is the bulk of the manuscript, needs to be enhanced. The Results section reads like a laundry list of the figures, so you might consider structuring your results around the scientific questions each figure answers rather than beginning each paragraph with "Figure N shows...". Given the current state of the Results section, I have a hard time knowing whether you think the model shows good agreement with some benchmark or plausible agreement with other model output. I also have no idea why you think the benchmarks you selected are the best ones to use (and whether they are sufficient), or which models you are trying to show that yours agrees with.

Fourth, I think that Fig. 4c merits a bit more attention. The authors contrast their FAIR model with the BEAM model by stating that FAIR includes some parameters for terrestrial uptake, rather than just marine uptake. Variations in terrestrial uptake largely drive variations in the airborne fraction, yet the FAIR model shows variability in the airborne fraction that is maybe 25% of the observations. So is FAIR really capturing climate-terrestrial uptake interactions? If variations don't show much semblance to

reality at shorter timescales, how can we trust the longer timescales? Given high profile recent papers that have tried to use interannual variability in the CO2 growth rate in ESM ensembles to constrain long-term temperature sensitivity of terrestrial uptake, I think this merits a bit more discussion.

The paper would also benefit from paying a bit more attention to details in the figures. For example, the legends for the figures are often incomplete and rely on information buried in the caption. Please help your reader by including this in the figure panels. The legend in Fig. 2 is incomplete (should include solid vs dash trendlines), as is the legend in Fig. 3 (should include red vs blue). There is no legend for Fig. 4. For Fig. 5, please consider using different colors since the purple and red are hard to distinguish. For Fig. 7 – I am not sure what are the brown dashed lines. It is possible that the purple shade and the brown shade are also too close.

In the revised paper, I think it is necessary to include more discussion of how do we know this simplified model is "good enough". The figures show comparisons against other, "wrong" models. Why is this model sufficient? Perhaps better discussion of the variables that comprise the model itself would and their physical significance/relationship to variables that exist in full-physics and full-BGC models would accomplish this, but the authors might also consider adding an additional section to the paper.

I would also like to see some discussion of how the FAIR model can be improved/extended in the future. Will variables simply be re-tuned when AR6 models come back with different emergent responses? Are there clear steps that would better emulate the physics, biology, and chemistry that govern the airborne CO2 feedbacks that can be added independent on next generation ESM simulations?

---

## Short Comment (SC1) · 20 Jul 2016

Contributed review of Atmos. Chem. Phys. Discuss., doi:10.5194/acp-2016-405

**A modified impulse-response representation of the global response to carbon dioxide emissions**
Richard J. Millar, Zebedee. R. Nicholls, Pierre Friedlingstein, and Myles R. Allen

Elisabeth Moyer / University of Chicago

This manuscript proposes a simple carbon cycle representation that involves a linear increase in integrated airborne fraction over time. Such a representation is useful for many purposes, including integrated assessment models that combine representations of the climate system and the economy. I assume that invited reviewers will cover the paper thoroughly so I will confine my remarks to a few concerns: the timescales and scenarios over which such a representation is useful, and how to make a valid comparison with the BEAM model described in Glotter et al 2014 (a paper on which I am a co-author).

*Timescales and scenarios*
The authors write: "We find that a simple linear increase in 100-year integrated airborne fraction with cumulative carbon uptake and global temperature change is both necessary and sufficient to represent the response of the climate system to CO2 on a range of timescales and under a range of experimental designs."

But, the airborne fraction does not increase linearly over long timescales in most realistic emissions scenarios. The figures in this paper focus on centennial timescales and on scenarios where $CO_2$ emissions are increasing at a constant growth rate. But if emissions decline, airborne fraction declines as well. (And over long timescales, cumulative airborne fraction continues to decline as the ocean comes into equilibrium.) This behavior can be shown by considering runs using the $A2^+$ emissions scenario, in which emissions rise as business-as-usual until 2100 and then drop linearly to zero by 2300 (Figure 1 below). The $A2^+$ scenario has been used with two intermediate-complexity Earth System Models (ESMs), UVic and CLIMBER-2. In these runs, airborne fraction rises initially but begins to drop as emissions slow, and continues dropping after emissions cease, reaching a final value of ~0.5. The authors of this manuscript need to be more specific about which conditions their model is able to capture.

[Figure]

**Figure 1**: Atmospheric $CO_2$ (expressed as GtC anomaly over year 2000 values) and airborne fraction in CLIMBER and UVic run with the $A2^+$ emissions scenario. In both models airborne fraction first rises and then falls. Airborne fraction is defined in two ways. The "cumulative" or "integrated" airborne fraction is the atmospheric anomaly above pre-industrial divided by the total past emissions. The "instantaneous" airborne fraction is the amount of $CO_2$ added or lost from the atmosphere in one timestep, divided by the emissions in that timestep. Instantaneous airborne fraction becomes negative in this scenario shortly before emissions reach zero, when net loss of $CO_2$ from the atmosphere exceeds emissions. (It is obviously not meaningful when emissions reach zero.) Cumulative airborne fraction reflects the entire emissions history; in this scenario nearly 9% of final cumulative emissions occur between 1880 and 2000.

I confess that I find the writing of this paper very confusing, and Figure 1d does seem to imply an decrease in airborne fraction over time in certain experiments, though followed by a subsequent increase. This decrease is not explained well, but is different from the long-term decrease that comes in realistic emissions scenarios when emissions slow. Figure 1 is introduced before either the model or the experiments has been described, which makes it hard to understand.

*BEAM comparison*
The second concern involves the comparison with the BEAM model. The version of BEAM used here is the default framework described in Glotter 2014; the authors write that BEAM was "run with parameters as given in Glotter et al. (2014), which are tuned for long time-scales". We see BEAM only as a framework that can capture the response of more complex models, and we would strongly prefer that parameters be chosen appropriate to the models being compared. Several of the parameters used in Glotter et al 2014 were in some sense placeholders, and we suggested that the users adjust them as needed to represent individual complex carbon cycle models. In retrospect we should have emphasized that need more clearly, and provided examples. It is evident from the manuscript figures here that default BEAM parameters do not capture short-term dynamics well. We are glad to provide code and discussion to ensure that the comparison of models is done appropriately.

While most of the parameters in BEAM have strong physical foundation, three are highly uncertain, those that relate the sizes and timescales of the reservoirs in the three-box models. These parameters seek to aggregate behavior globally that is not well-measured even locally. They are 1) $\delta$, the ratio between the upper and lower oceans, represented as distinct reservoirs; 2) $k_d$, the exchange time between them; and 3) $k_a$, the exchange time between atmosphere and upper ocean. In the presentation of BEAM in Glotter 2014 we (in retrospect, unwisely) left these values as the fairly arbitrary choices of Bolin and Eriksson 1959 ($\delta = 50$, $k_d = .05$, and $k_d = .2$), and suggested (in appendix A.2) that users should adjust them as needed to best represent more complex ocean models. We had assumed that the primary use for BEAM would be in simple Integrated Assessment Models that consider long timescales and require relatively crude representations of the physical climate system. The Bolin and Eriksson values seemed acceptable for this purpose, as resulting temperature differences between BEAM and the ESMs are no more than 0.23 K in the first 100 years, and thereafter the two ESMs bracket BEAM temperatures.

However, the Bolin and Erikkson parameters describe a relatively shallow mixed-layer upper ocean, equivalent to about 75 meters depth, with a rapid exchange timescale of $1/k_d = 20$ years. In this configuration, even very small $CO_2$ uptake can alter the acidity of the shallow mixed layer substantially, so that the parameter B, which describes the ratio of dissolved $CO_2$ to total inorganic carbon, rises strongly. That rise means that very little $CO_2$ uptake is required to keep the mixed layer in equilibrium with the atmosphere, and atmospheric $CO_2$ drawdown is driven predominantly by transfer of inorganic carbon to the deep ocean. The result is a very high initial instantaneous airborne fraction, which drops only when emissions growth slows.

BEAM output can be readily matched to that of more complex climate models by adjusting the ratio of ocean reservoir volume $\delta$ and their exchange timescale $k_d$. (The upper ocean exchange timescale is sufficiently fast that it is less relevant at centennial timescales.) Figure 2 below shows $CO_2$ and airborne fraction under the A2[+] scenario, as in Figure 1, but here we compare output from UVic with that of BEAM with a variety of parameter settings: the Bolin and Eriksson value, a global optimized value to best capture UVic behavior, and output with $k_d$ optimized for various values of $\delta$. Results show that UVic is best emulated with a larger mixed layer volume and a longer exchange timescale. With this

representation a given amount of $CO_2$ uptake produces smaller acidity changes, so that more uptake is required to bring the upper ocean to equilibrium with the atmosphere.

[Figure]

**Figure 2**: Optimizing the fit of BEAM to UVic output by adjusting the parameters $\delta$ and $k_d$ (The exchange timescale $\tau d = 1/k_d$.) The very high instantaneous airborne fractions noted by the authors here are the result of an upper ocean mixed layer that is too shallow. The optimized parameters (red) describe a very deep mixed layer with a long exchange timescale. Adding a linearly declining land sink (orange, indistinguishable from red) produces an equally faithful reproduction of UVic output but with a more realistic mixed layer.

Note that the optimized parameters are still not the most physically reasonable, as the optimization exercise forces an ocean-only model to reproduce atmospheric $CO_2$ from ESMs that include land carbon sinks as well. That is, in BEAM the ocean is forced to account for all the uptake of the present-day land sink as well, which is believed to be comparable in magnitude to the ocean sink. The authors here state that BEAM cannot be compared to models that include land sinks, but it is a trivial to add one. Of course, the long-term evolution of the land sink is highly uncertain and differs even in sign in more complex models, so the exercise is most useful for studies that focus on the short term. As a demonstration, we have added a land sink that begins at 2.5 GtC/year in the present day (year 2000) and declines linearly to zero in 300 years and calculated optimized parameter values. The exercise yields slightly larger $\delta$ and $k_d$ and again matches UVic $CO_2$ trajectories well (Figure 2). This formulation may be best to use when comparing across emission scenarios, since the more physically rational the representation, the better able the model will be to capture responses to differing emissions trajectories. Again, we are happy to provide code if that is helpful.
* * *
**Additional confusions**

I am confused about the author's definition of "cumulative uptake" and "cumulative airborne fraction" in Figure 1. In this figure BEAM output is shown as beginning with ~300 ppm and zero cumulative uptake. But the initial conditions suggested in Glotter et al 2014 begin BEAM after historical emissions from 1800-2000, so that starting atmospheric CO2 is over 380 ppm, and substantial emissions and uptake have occurred already.

In addition, given those initial conditions, the starting "cumulative airborne fraction" is ~0.5 and rises only slowly over time even when ocean uptake is small and instantaneous airborne fraction is high. Here the cumulative airborne fraction is shown as reaching 0.9 nearly immediately.

Finally, I was confused by statements implying that different emissions scenarios can be captured by a model that represents airborne fraction as a function of cumulative emissions (and temperature). Again the writing is confusing and I may have misunderstood, but airborne fraction seems quite sensitive to the emissions scenario (Figure 3 below). It seems that a figure is needed to explicitly validate this assertion.

[Figure]

**Figure 3**: Instantaneous and cumulative airborne fraction in BEAM under three emissions scenarios, A2[+] and scenarios with A2[+] emissions doubled or halved. The version of BEAM is that optimized for reproducing UVic, with an assumed land carbon sink beginning at 2.5 GtC/year and linearly declining over 300 years. Airborne fraction is dependent on emissions scenario in both short and long terms. The vertical lines in the right panel represent the period when emissions have ceased but atmospheric $CO_2$ slowly declines over thousands of years.

---

## Author Comment (AC2) · 21 Jul 2016

As an example of the ability of the FAIR model structure to reproduce the ESMs and EMICs shown from Joos et al (2013) we here include an example figure of model comparison. We would incorporate a version of this figure in a revised version of the manuscript.

In figure 1 (of this comment) we fit parameters in FAIR to individual models in a two stage process. Firstly, we fit parameters to minimise the total sum of squares residual in both the PD100 and PI100 experiments for the multi-model mean. Free parameters used are the coefficients partitioning the emissions into different timescale carbon pools ($a_i$ in equation 1 of the discussion paper), the $r_0$ parameter and proportional variations in the $r_C$ and $r_T$ parameters (therefore maintaining a fixed ratio of $r_T/r_C = 225$GtC/K as in the discussion paper) whilst keeping the $\tau_i$'s (timescales) fixed at the AR5-IR model values. The temperature timeseries is taken as given in the model output to focus the fitting exercise solely on the carbon-cycle behaviour. The best fit to the multi-model mean (dashed black line in figure 1 of this comment) selects $a_i$ parameters that are different to the AR5-IR parameters due to the differing model formulation.

Now fixing the $a_i$ parameters at these new values from the multi-model mean fit, we then attempt to fit the individual models from Joos et al (2013) using just two degrees of freedom: $r_0$ and proportional variations of both the $r_C$ and $r_T$ parameters (again leaving the ratio fixed). As can be seen in figure 1 of this comment, a diverse set of model responses can be emulated within the FAIR structure by changing just two important degrees of freedom in the parametrisation of changes in $iIRF_{100}$, namely the pre-industrial magnitude of the carbon-cycle response ($r_0$) and the total size of the feedbacks. A single set of parameters of the FAIR model successfully captures the response in both the PD100 and the PI100 experiments for individual models.

We hope an expansion and discussion of this figure in a revised version of the manuscript would address several of the comments of both reviewer 1 and reviewer 2 with regard to model evaluation and comparison to ESMs.

[Figure]

[Figure]

**Fig. 1.** Fits to the EMICs and ESMs from Joos et al (2013). Solid lines represent the raw model data and the dashed lines the FAIR fits in each case. The lines are coloured by their iIRF100 values.

[Figure]

---

## Author Response (AR1)

**Response to reviews and short comments**

Responses to the reviews and the short comment received are included below in the form of in-line response for clarity. A list of changes is not included (and most likely not of any use) given of the restructuring in response to one of the reviews. Specific changes can be identified from the tracked changes version of the manuscript included here.

**Response to Reviewer 1**

Despite the reviewers recommendation we believe that the points raised can be relatively simply dealt with and would hope the making the changes outlines through the in-line responses to the specific points raised (red, italic) below would change the reviewers recommendation.

*The authors motivate the need for their modified impulse response function model by claiming "This extension is necessary because the use of a state-insensitive impulseresponse model cannot simultaneously reproduce the relationship between emissions, concentrations and temperatures seen over the historical period and the projected response over the 21st century to both high-emission and mitigation scenarios estimated from more complex models." (p. 2, lines 23ff). While it is true that the Joos et al. (2013) model is not accounting for a state dependence, it got not really clear to me from the paper whether this unability of 'simultaneous reproduction' is true or not.*

Figure 4 of the original manuscript shows that the AR5-IR model (red) fails to reproduce observed concentrations over the historical period when integrated with historical emissions (panel a) and fails to reproduce historical emissions when emissions consistent with historical concentrations are derived from the model (panel b). A substantial error of nearly 30 ppm is seen in figure 4a, representing a large difference from the correct climate state. We also show in figure 5 how the AR5-IR model fails to produce similar behavior to the MAGICC model under the RCP8.5 and RCP2.6 scenario. The MAGICC model is approximately consistent with the response of the ESMs in CMIP5. We agree with the reviewer that this point could be made more prominently and have emphasized this point more in the revised version of the manuscript.

*"it is encouraging that the FAIR model shows a close correspondence with a well-known and well-used simple model [=MAGICC] that has been used extensively to emulate the response of ESMs" (p. 8, lines 7f) ('encouraging' is nice but not convincing).*

We agree with the reviewer that agreement between two simple models of the climate system is only encouraging and is not, and cannot ever be, a sufficient evaluation of model skill. This is the point we wished to convey here and deliberately chose not to use a stronger word precisely for this reason. We have clarified why we compare to MAGICC (in order to produce a single set of emissions for the RCP scenarios that all simple models can be driven by) in the revised manuscripts discussion of this figure.

*In particular, it is not well specified which model simulations are meant to be emulated by FAIR.*

We intend for the FAIR model to reproduce features of the climate response to $CO_2$ as shown in ESMs (e.g. p. 2, l. 19-20). We have often referred to "more complex/comprehensive models" in the text to indicate that we are attempting to emulate behavior simulated by both ESMs and EMICs (e.g. p. 2 l. 4) but we agree that this wording is unhelpfully ambiguous, particularly in the abstract, and has been rectified in a revised version of the manuscript.

The reviewer raises several points about the method of comparison of our model to other simple models and ESMs. We explicitly didn't set out to do a full model tuning exercise, in which the consistency of the FAIR model with other ESM simulations is evaluated through defined metrics, as this paper was intended to be a simple demonstrate on how with this parameter set (which hasn't been formally derived in any way) approximate consistency with the historical record and behavior of ESMs could be attained. A full parameter fitting exercise would be worthwhile with the FAIR model, but as there are fewer free parameters within FAIR than constraints, the output of such a tuning procedure would depend on exactly which consistencies are of most interest to the users of the model and would therefore need to be undertaken on a case-by-case basis depending of the desired usage. We have added a new figure (Figure 4 of the revised manuscript that shows a possible tuning exercise with the FAIR model to reproduce

particular aspects of ESM behaviour.

*the results of several impulse experiments are discussed in section 3, resulting in questionable claims on the quality of their new model like "consistent with corresponding ratio in the most detailed ESMs" (p. 6, lines 23f)* (what means 'most detailed'?)

We intended consistency to refer to the quantative statement in the following clause, namely, "with its value of 36 years within the 34-47 years range of the ESMs". We agree with the reviewer that the word 'detailed' adds no extra information, introduces confusion and has been removed in the revised version of the manuscript. We have added ranges of the pulse responses in ESMs from Joos et al (2013) onto the orginal figure 3 to display this comparison visually.

*"the FAIR model can capture the dependence of the pulse-response on pulse size" (p. 6, line 28) (what means 'capture'? In comparison to what?)*

The black lines showing the range of response from the Joos et al (2013) models to the pulse-response experiments figure (Figure 3 of the revised manuscript) allow this dependency and the ability to reproduce it to be seen.

**Response to Reviewer 2**

We enclose responses to the points raised by reviewer 2 (red) in-line below…

First, the authors must provide a better literature review that contains brief descriptions of the other models to which their FAIR model is compared, including the IPCC simple model and the BEAM model, which is introduced rather abruptly in the results section.

We provide a lengthier introduction to the simple carbon cycle models in the revised version of the manuscript

At the end of the introduction, the authors propose to "extend" on the IPCC-AR5 model, but I don't think they have fully introduced this model, or described why they think it is deficient.

We introduce this model more thoroughly to set the modified version of this model (FAIR) in better context in the revised manuscript.

Second, the authors must provide a better description of their own FAIR model. The time constants should be better described, the carbon reservoirs used in the model should be named, and all the variables used in equations 1-5 should be described and the appropriate units should be listed.

The impulse-response formulation of the AR5 and FAIR models are empirical models that are based on the fits to the response of more complex models as done in Joos et al. Therefore the specific timescales and carbon reservoirs in the model do not necessarily have simple physical interpretations and as such it is impossible to assign specific names to them. This point is added to the revised version of the manuscript. However, approximate correspondence exists between physical processes and decay timescales, which is now commented on in the revised version of the manuscript (table 1 – which documents parameters and their units).

Third, the discussion of the figures, which is the bulk of the manuscript, needs to be enhanced. The Results section reads like a laundry list of the figures, so you might consider structuring your

results around the scientific questions each figure answers rather than beginning each paragraph with "Figure N shows...". Given the current state of the Results section, I have a hard time knowing whether you think the model shows good agreement with some benchmark or plausible agreement with other model output. I also have no idea why you think the benchmarks you selected are the best ones to use (and whether they are sufficient), or which models you are trying to show that yours
agrees with.

We agree that the results section could have been better structured around the scientific questions raised by each figure. We have re-organized and rewritten the results section as appropriate and have included ESM/EMIC data in the figures where appropriate to be clear on the success and limitations of our model evaluation.

Fourth, I think that Fig. 4c merits a bit more attention. The authors contrast their FAIR model with the BEAM model by stating that FAIR includes some parameters for terrestrial uptake, rather than just marine uptake. Variations in terrestrial uptake largely drive variations in the airborne fraction, yet the FAIR model shows variability in the airborne fraction that is maybe 25% of the observations. So is FAIR really capturing climate-terrestrial uptake interactions? If variations don't show much semblance toreality at shorter timescales, how can we trust the longer timescales? Given high profile recent papers that have tried to use interannual variability in the CO2 growth rate
in ESM ensembles to constrain long-term temperature sensitivity of terrestrial uptake, I think this merits a bit more discussion.

As FAIR (and the other impulse-response models considered in this paper only represent the externally-forced response of the climate-carbon-cycle system they are likely to be structurally unable to reproduce the observational variations in airborne fraction (which are likely to be driven by internal variability in a large fraction). We have added this to the paper and indicated that more complex models are required to understand the drivers of these variations and their implications for future carbon-cycle response.

The paper would also benefit from paying a bit more attention to details in the figures. For example, the legends for the figures are

often incomplete and rely on information buried in the caption. Please help your reader by including this in the figure panels. The legend in Fig. 2 is incomplete (should include solid vs dash trendlines), as is the
legend in Fig. 3 (should include red vs blue). There is no legend for Fig. 4. For Fig. 5, please consider using different colors since the purple and red are hard to distinguish. For Fig. 7 – I am not sure what are the brown dashed lines. It is possible that the purple shade and the brown shade are also too close.

We have improved the presentation of our figures in the revised version of the manuscript.

In the revised paper, I think it is necessary to include more discussion of how do we know this simplified model is "good enough". The figures show comparisons against other, "wrong" models. Why is this model sufficient? Perhaps better discussion of the variables that comprise the model itself would and their physical significance/relationship to variables that exist in full-physics and full-BGC models would accomplish this, but the authors might also consider adding an additional section to the paper.

We are not exactly sure what the reviewer is asking for here, but we interpret 'how do we know the model is good enough/sufficient' as asking whether this model structure is capable of spanning the range of responses seen in ESMs. We hope the addition of the new figure and the inclusion of ESM/EMIC data on the figures addresses this point by showing that the FAIR model can act as an effective emulator of the range of response in the Joos et al models with perturbations to only a subset of the parameters within the FAIR model.

I would also like to see some discussion of how the FAIR model can be improved/ extended in the future. Will variables simply be re-tuned when AR6 models come back with different emergent responses? Are there clear steps that would better emulate the physics, biology, and chemistry that govern the airborne $CO_2$ feedbacks that can be added independent on next generation ESM simulations?

We do see FAIR as a model emulator framework in which

the parameters can be tuned to reproduce CMIP5/CMIP6 behaviour whilst maintaining the physically understood dependencies of the response on background conditions and pulse size. We have revised the manuscript to reflect this perspective more clearly including a new figure in which we emulate the response of Joos et al models to pulses emissions of $CO_2$ with a set of parameters appropriate for both present-day and pre-industrial pulse responses.

**Response to SC1 by Elisabeth Moyer**

Professor Moyer raises some interesting points in her short comment on our ACPD paper, which influenced our revisions and has enhanced the revised paper. We address some of the main points raised (red,italic) in the comment in-line here:

*The authors write: "We find that a simple linear increase in 100-year integrated airborne fraction with cumulative carbon uptake and global temperature change is both necessary and sufficient to represent the response of the climate system to CO2 on a range of timescales and under a range of experimental designs." But, the airborne fraction does not increase linearly over long timescales in most realistic emissions scenarios.*

We parameterize the *integrated* airborne fraction as linearly increasing with warming and cumulative uptake. This does not imply that the instantaneous airborne fraction would be expected to increase linearly with time. Indeed the FAIR model displays a similar temporal evolution of the instantaneous airborne fraction under the A2+ scenario as shown by the UVic EMIC (shown in Figure 1 of this comment which shows the same data as Figure 1 of SC1 with the FAIR model). After emissions stop in this scenario (2300) atmospheric concentrations decay away with four characteristic timescales back toward the pre-industrial equilibrium. The most relevant timescales for the several hundred years is associated with equilibration of the full-depth of the ocean. We see no conflict between the FAIR formulation and these features of the UVic/CLIMBER models under the A2+ scenarios.

[Figure]

Figure 1: The response of the FAIR model to the A2+ emissions scenario. The dashed lines correspond to the right hand axis.

*Figure 1d does seem to imply a decrease in airborne fraction over time in certain experiments, though followed by a subsequent increase. This decrease is not explained well, but is different from the long-term decrease that comes in realistic emissions scenarios when emissions slow*

Figure 1d shows the *cumulative* airborne fraction, and as highlighted in SC1, the cumulative airborne fraction represents the integrated evolution of the instantaneous airborne fraction over pervious years. The initial decrease in cumulative airborne fraction, followed by a subsequent increase, demonstrated by the FAIR model is a feature of the response of many ESMs under a 1%/yr increasing $CO_2$ scenario. Figure 2 of this comment shows a new version of Figure 1 of the original manuscript that shows pastel-coloured lines associated with the ESMs analysed in Arora et al. (2013) under a 1%/yr $CO_2$ increase scenario. The initial decrease in cumulative airborne fraction followed by subsequent increase can be understood in terms of the saturation of carbon sinks. If atmospheric anomalies of carbon decay with invariant timescales (as in the AR5-IR model case), then instantaneous airborne fraction remains constant in time, which necessarily means that cumulative airborne fraction must decline in time (as emissions from previous years decay further, so the cumulative fraction of the emitted carbon continually decays from the instantaneous airborne fraction), this behaviour is shown by the AR5-IR model in figure 1d of the main text and figure 2 here. However, if carbon sinks become saturated, the instantaneous airborne fraction would be expected to increase with time (this is represented in the FAIR model by increases to the decay timescales through the parameterised increase in iIRF100). As more recent emissions (which are of increasing magnitude under the 1%/yr scenario) have a higher instantaneous airborne fraction, the cumulative airborne fraction decrease stops and then begins to increase again as this accelerating saturation becomes the dominant effect. We have provided a more in-depth discussion of this in the revised manuscript to help better communicate the results shown in figure 1d.

[Figure]

Figure 2: As for figure 1 in the main text, but new coloured lines included correspond to the response of the ESMs analysed in Arora et al (2013) under the 1%/yr CO2 increase scenario.

*Figure 1 is introduced before either the model or the experiments has been described, which makes it hard to understand.*

We begin our discussion of Figure 1 with the statement "Figure 1 shows the response of the FAIR model (blue) described in section 2 under the three experiments described above." so we do not believe that this to be the case. However, our discussion of Figure 1 evidently needed to be clearer and we believe we have improved this in our revised manuscript to hopefully make it clearer for readers to follow.

*We see BEAM only as a framework that can capture the response of more complex models, and we would strongly prefer*

*that parameters be chosen appropriate to the models being compared.*

We used the default parameter settings from Glotter et al. (2014) as those were the only consistent set that were documented in the paper. However, we agree that in general simple models should be tuned to specific more complex models. The version of BEAM given in Glotter et al (2014) is clearly a good approximation of the response of the UVic and CLIMBER EMICs under the A2+ scenario as shown in the figures of Glotter et al (2014). We feel that documenting a method for tuning the BEAM model, and the results of the results of that process, would be a worthy investigation in its own right but not something that would be possible within the constraints on this paper here, where the focus is on the FAIR model. On reflection, as our main point in the paper is demonstrating the need to include state-dependence within impulse-response carbon-cycle models, we feel that inclusion of BEAM as a comparison model for just one figure in the paper (as was the case in the original manuscript) does little to enhance the paper and maybe adds unnecessary baggage to communication of its core message. We have therefore removed explicit discussion and comparison to BEAM in the revised manuscript and instead focus solely on comparisons to state-independent impulse-response climate-carbon-cycle models. We would however be interested in doing an extended comparison of FAIR with BEAM as a separate study, including tuned versions of the BEAM parameters, which we would be delighted to collaborate with Professor Moyer on in the future.

*We had assumed that the primary use for BEAM would be in simple Integrated Assessment Models that consider long timescales and require relatively crude representations of the physical climate system. The Bolin and Eriksson values seemed acceptable for this purpose, as resulting temperature differences between BEAM and the ESMs are no more than 0.23 K in the first 100 years, and thereafter the two ESMs bracket BEAM temperatures*

We would argue that because of the economic practice of discounting future damages in conventional integrated assessment activities, the response of a carbon-cycle model to a pulse emission of carbon over the first 100 years is of most importance for Integrated Assessment Models.  Even at moderate discount

rates of about 2.5%/yr the weighting for climate damages driven by physical changes are only 0.08 of today's weighting. Whilst this relative over-weighting of the multi-century scale response may be an undesirable feature of the economic paradigm, it therefore currently remains most important to correctly represent the response of the climate system over the first 100 years in a physical model that will be used as part of Integrated Assessment Models and particularly in calculations of the social cost of carbon. This is why we chose to structure the FAIR model in terms of an explicitly parametrised iIRF100 to allow this timescale of the model's response to be mapped to the behavior of a range of ESMs as transparently as possible.

*I am confused about the author's definition of "cumulative uptake" and "cumulative airborne fraction" in Figure 1. In this figure BEAM output is shown as beginning with ~300 ppm and zero cumulative uptake. But the initial conditions suggested in Glotter et al 2014 begin BEAM after historical emissions from 1800-2000, so that starting atmospheric CO2 is over 380 ppm, and substantial emissions and uptake have occurred already. In addition, given those initial conditions, the starting "cumulative airborne fraction" is ~0.5 and rises only slowly over time even when ocean uptake is small and instantaneous airborne fraction is high. Here the cumulative airborne fraction is shown as reaching 0.9 nearly immediately.*

Cumulative uptake is defined as the total amount of the carbon previously emitted into the atmosphere that has been removed from the air by the carbon-cycle system. Cumulative airborne fraction is the fraction of the previously emitted carbon that still remains in the atmosphere at a point in time. We have tried to make this clearer in the introduction of the FAIR model in section 2 of the revised manuscript. The definitions used in the manuscript are consistent with those used in SC1. The confusion mentioned above may arise is part because Figure 1 represents scenarios that begin from pre-industrial conditions with concentrations increasing by 1%/yr, 0.5%/yr and 2%/yr. Glotter et al (2014) also suggested a pre-industrial initial condition for BEAM (Table 4, Appendix A.3) which corresponds to a pre-industrial concentration of 280ppm. It is these initial conditions that were used in Figure 1 of the original manuscript.

*Finally, I was confused by statements implying that different emissions scenarios can be captured by a model that represents airborne fraction as a function of cumulative emissions (and temperature). Again the writing is confusing and I may have misunderstood, but airborne fraction seems quite sensitive to the emissions scenario (Figure 3 below). It seems that a figure is needed to explicitly validate this assertion.*

We agree that the response of a carbon-cycle-climate model is dependent on the emissions scenario. The long dashed and short dashed lines in figure 2 of this comment show that the FAIR model can capture dependencies of the response on the emissions scenario, similarly to as shown for two ESMs in Gregory et al (2009) (Figures 4, 5, 6 of that paper). We have additionally reproduced Figure 3 of SC1, with the FAIR model (Figure 3 of this comment) that demonstrates that the scenario dependencies of the BEAM model under variants of the A2+ emissions scenario can be similarly represented with FAIR.

[Figure]

Figure 3: As in Figure 3 of SC1 but for the FAIR model. The red line represents a 2xA2+ emissions scenario, the black the standard A2+ emissions scenario and the green a 0.5xA2+ emissions scenario.

[revised manuscript text omitted]

---

## Author Response (AR2)

**Reviewer response and marked-up manuscript**

We include a response to the reviewer's comments and a marked up version of the changed manuscript below. We do not include a list of changes due to the comprehensive restructuring of the manuscript as requested by the reviewer.

**Response to reviewer**

We thank the reviewer for a very thorough and useful review of our manuscript, which has greatly improved the manuscript. We answer the reviewer's specific points in line below.

1. Even after reading the paper, I do not really understand the last sentence of the Abstract.

We have removed the final sentence of the abstract in order to make it more focused on the core messages and analysis in the paper.

2. I found the Introduction generally good, but maybe the need for the study could be made clearer by noting that the results from Joos et al. (2013) clearly show that: (1) the airborne fraction for a given pulse size depends upon the background state of the atmosphere; and (2) the airborne fraction for a given background state of the atmosphere depends upon the pulse size. Conversely, the standard IRF is state independent...

We have included these points more prominently in the introduction.

3. Page 2, line 16. Which "simple climate-carbon cycle models"? MAGICC? Standard IRF? Depending upon the answer, stating that these models have not been "evaluated in terms of their pulse-response behaviour" could be wrong. Similar comment for page 2, line 26.

We have been clearer on this point in order to rule out those simple models that were included in the Joos et al. 2013 paper and now give DICE as a specific example of an untested IAM carbon cycle model.

4. Page 2, line 25. In my opinion Davis and Socolow (2014) did not really evaluate the "required energy-system transitions that are needed to limit warming to below particular thresholds"; they rather estimated emissions from existing infrastructure.

We have updated our use of the reference to better reflect the content of this paper.

5. Page 3, line 15. The 4-exponentials IRF is for mathematical convenience and does not really correspond to the time dynamics of actual mechanisms, so the

"processes" provided in Table 1 are just 'guiding analogies'. This should be noted here instead of below and made clearer.

Changed in the revised manuscript.

6. Page 3, line 18. Replace "i = 1, 4" with "i = 1, ..., 4" or "i = 1–4" or "i = 1 to 4".

Changed in the revised manuscript.

7. Page 3, line 25. Mention that the authors decided to give a finite ($1\times10^6$) value to $\tau_0$, as this differs from Myhre et al. (2013). Also mention (here or elsewhere) that for PI-IR, the $a_i$ come from Table S2 of Joos et al. (2013).

Included in the revised manuscript. We have updated the PI-IR parameters to fit the multi-model mean of the Joos et al PI100 experiment, the parameters of which were not included in the original paper. We have stated this is the provenance of the PI-IR parameters used in the revised text.

8. Page 3, line 25-28. Defining ECS and TCR (and TCRE when mentioned much later in the paper) would probably be worthwhile.

Included in the revised manuscript.

9. Page 3, lines 25-28. The link between the $c_j$ and both ECS and TCR is not clear. The reference to Millar et al. (2015) does not really clarify this point as the latter study presents $c_j$ and $d_j$ in their Supplement only, and under a mathematical form that differs from Eq. (3) considered here. To address this and other comments below, the authors need to add an Appendix or a Supplement in which they: (1) clearly show the mathematical link between the $c_j$ and $d_j$ as appearing in Eq. (3) and both ECS and TCR; and (2) clearly explain how they obtained $c_1 = 0.46$ and $c_2 = 0.27$.

We have provided information about the link between ECS/TCR and the model parameters with equations 4 and 5 in the revised manuscript. We do not believe this requires an appendix or supplement and hope that detail provided is sufficient to allow readers to understand how to invert the equations to solve for $q_1$ and $q_2$.

10. Page 4, Table 1. Provide the units for $c_1$ and $c_2$.

Units provided. We have changes the notation for $c_1$ and $c_2$ to $q_1$ and $q_2$ to avoid having both upper and lower case "c" variables within the paper.

11. Page 5, line 10. iIRF is not the "average airborne fraction over a period of time", but the product is this average fraction with the length of the integration period.

Corrected in the revised manuscript.

12. Page 5, line 21 to page 6, line 2. The text is poorly structured; I think the following order would help. First, explain how FAIR uses Eqs. (4) to (6) at each time step, along with Eq. (3) for the temperature. Second, explain how were determined the specific values of r_0, r_C, and r_T (was it simply through trial and error? until finding what?). Third, state that these values work well but could be tuned even more (i.e., the text that currently appears on lines 21-27). Fourth, address the iIRF_100 > 100 years issue (would it really occur in the runs if iIRF_100 had not been limited to 95 years? if yes, the authors need to discuss the implications of this issue later on in the text).

We have corrected the order of explanation as suggested by the reviewer. We have also expanded on the selection of r_0, r_C and r_T and the effect of a maximum iIRF100 within the text.

13. Page 5, line 29: "This means the iIRF_100 is only exactly reproduced [...]". Why?

We have added to the text to explain that this expression is only valid if \alpha is assumed to be invariant in time, a limit that would be reached for an arbitrarily small pulse.

14. Page 6, line 3. Adding one or two new methodological subsection(s) is required to clearly explain the simulations performed, how FAIR parameters were modified (uncertainty analyses), give the sources of input data (RCP, etc.), etc. Much of the text from the Results should be transferred here and expanded. Below, I refer to these new subsection(s) as "2.3".

We agree that a methods section would aid comprehensibility and readability of the manuscript and have included one as requested.

15. Page 6, line 3. Results are discussed as they are presented, which I think is appropriate in this paper. Therefore, the section should probably be named "Results and discussion".

Changed in the revised manuscript.

16. Page 6, lines 11-13. These two sentences are not necessary.

Removed in the revised manuscript.

17. Page 6, lines 13+. Two undiscussed elements stroke me when looking at Fig. 1. First, one would expect PI-IR to end up with lower atmospheric CO2 than historical observations (because PI-IR CO2 sinks work with pre-industrial

efficiency throughout) but this is not the case; why? I think this is because PI-IR was obtained by Joos et al. (2013) under a pulse of 100 GtC, whereas historical annual emissions were much lower and therefore initially had less impact on CO2 sinks efficiency. The authors should provide this explanation (if they agree with it) as it addresses the issue and strengthens their point about the inadequacy of state-independent IRF model. Second, PI-IR CO2 sinks are less efficient than FAIR CO2 sinks until about year 2000 (Fig. 1c). This seems mathematically impossible when looking at parameter values in Table 1 and the different equations... unless \alpha in FAIR has a value < 1. I think the authors should explain this here, and also give in the Methods the initial value of \alpha (about 0.16, right?) when C_acc and T are still equal to zero (i.e., when iIRF_100 is equal to r_0). The way FAIR is introduced, I initially thought \alpha would always be > 1 and got confused.

We agree with the reviewer regarding their explanation of the PI-IR curve in Figure 1a (along with the inability of the temperature independent AR5-IR and PI-IR models to capture temporary reductions in iIRF100 due to naturally forced cooling) and have explicitly stated this in the revised manuscript. We have included the value of \alpha in the pre-industrial state for reference in the revised manuscript.

18. Page 6, lines 16-17. "The AR5-IR displays a too large [...]". This sentence is a poor description of Fig. 1c, as no single model is really "consistent with the observations". The authors can only state that AR5-IR is always higher than FAIR and that both are much more stable than observations.

We have removed these phrases and comparisons from the revised manuscript.

19. Page 7, line 6. Mentioning the social cost carbon one time in the Introduction is OK, but coming back to this concept throughout the paper seems out of place (has the paper been written for another journal?) and pointless (a model of CO2 dynamics needs to give good results to be useful for any application, not just the social cost of carbon); please remove. Similar comment for page 2, line 13; page 6, lines 15-17; and page 13, line 13.

Removed in the revised manuscript.

20. Section 3.1. The text should refer to the results in Fig. 1d, Fig. 2c, and Fig. 2d or these panels should be removed. With a current total of 28 Figure panels, less would probably be better.

All panels shown are now discussed in the text. We have also streamlined the number of figures in the text to help communicate the main points of the paper.

21. Page 7, lines 15-19. This is Introduction-type text, not for the Results.

Now included in the introduction.

22. Section 3.2. The authors often mention iIRF_100, but this variable is not shown in the Figures. The authors should present airborne fraction results instead or add a Table with iIRF_100 results.

A table of iIRF100 results is now included.

23. Page 7, lines 20-26. The majority of this text belongs to 2.3, along with the explanations about how "fully-coupled", "biogeochemically-coupled", and "radiatively-coupled" results were obtained. I also suggest removing Fig. 3a, which is more 'methodological' (i.e., diagnosed emissions required to reach a particular CO2 level) and not really interesting in itself.

This material is moved to the new methods and the figure panel removed.

24. Page 7, lines 31-32. All models show a rapid temperature increase followed by a relatively stable value, not just the "fully-coupled" FAIR model.

Changed in the revised manuscript.

25. Page 8, lines 1-3. Cumbersome sentence; please rephrase.

Phrasing changed in the revised version of the manuscript.

26. Page 8, lines 18-26. The majority of this text belongs to 2.3, where the decision to maintain the same ratio between r_T and r_C needs to be justified. But in fact, I even suggest removing Fig. 4 from the paper as I do not believe it adds much value.

Text moved to methods section and the choice of fixed r_T : r_C ratio (as the fully-coupled PD100/PI100 experiments do not constrain the balance between r_T and r_C) is discussed. We choose to keep figure 4 as we believe that it shows that FAIR is capable to simulating the carbon-cycle responses in the full range of ESM and EMIC models from Joos et al in both the PD100 and PI100 experiments using just a single set of parameters for each model, an important demonstration of versatility for policy relevant use where spanning ranges of ESM responses is important. This analysis was also explicitly asked for by a reviewer in the previous round.

27. Page 8, line 27. Actually, the FAIR model is able to "successfully capture much of the response" only for the well-behaving models (i.e., no major year-to-year variability); this should be noted.

We note this caveat about not being able to simulated complex model inter-annual variability.

28. Page 8, line 31 to page 9, line 8. The majority of this text belongs to 2.3, with possibly some elements to the Introduction.

Text moved in the revised version of the manuscript.

29. Page 8, lines 5-12. The authors apparently misunderstood Zickfeld and Herrington (2015). The issue with the results of Ricke and Caldeira (2014) is not so much that they did not "account[] for feedbacks on the carbon cycle and fail[ed] to capture the plateau of CO2-induced warming" as that they did not account for the effect of the pulse size on the shape of the temperature response (because they used a state-independent IRF model): for very large pulses, there is no longer an early warming peak followed by a plateau. FAIR is able to capture this change of shape from small pulses leading to an early warming peak (Fig. 8d) to large pulses without an early warming peak (Fig. 5a), whereas the standard IRF model is not (Fig. 5a). Although this outcome further illustrates the scientific value of FAIR, this change of shape occurs for pulses > 1000 GtC (also see Zickfeld and Herrington, 2015) and is therefore of little practical relevance for real emission scenarios. Given the amount of results presented, I thus suggest removing Fig. 5 from the paper as I do not believe it adds much value. If the Figure is kept, the text describing its results should be made accurate.

We have revised our discussion of Zickfeld and Herrington to focus the discussion on the dependence of maximal warming on the pulse size and have attempted to clarify and tighten the discussion around figure 5.

30. Page 9, lines 14-31. The majority of this text belongs to 2.3, where the 'stopping rule' for the different %/yr simulations should be given (until quadrupling initial CO2?). The long sentence on lines 23-26 adds little value.

Text moved to the methods section and stopping rule specified. We have moved the sentence from page 9 line 23-26 to the conclusions. We choose to retain this sentence as we do believe the integrated analysis of mitigation and solar radiation management requires a model that correctly distinguishes the effects of warming and carbon on the carbon-cycle and that this is a worthwhile point to make. However this point perhaps fits better in the conclusion section.

31. Page 10, line 3. Why does a "constant airborne fraction necessarily give[] an approximately quadratic increase" and what is "approximately quadratic" (e.g., an exponent of 1.8)?

This statement, which the reviewer correctly identified as confusing has been removed from the revised manuscript.

32. Page 10, lines 6-8. Cumbersome sentence; please rephrase.

Rephrased in the revised manuscript.

33. Page 10, lines 8-10. Specify that these results are for "radiatively-coupled" experiments.

Changed in revised manuscript.

34. Page 10, lines 10-14. Cumbersome sentence; please rephrase.

Rephrased in revised manuscript.

35. Page 10, line 15. The "cumulative airborne fraction" could be more clearly defined: it is the fraction of all past emissions that are still in the atmosphere.

Changed in revised manuscript.

36. Page 10, lines 29-34. This is Introduction-type text, not for the Results.

Moved in revised manuscript.

37. Page 11, lines 1-5. Much more details about what was done and how it was done (i.e., which was the range of c_j values used) must be provided in 2.3 and possibly the new Appendix/Supplement. Figs. 7a and 7b should also be better introduced.

We now provide the range of q_j used to sample the TCR and ECS range. As mentioned above, we also provide more detail in the model description about how to link TCR/ECS and q_j. We try to introduce figure 7a and 7b better in the revised manuscript.

38. Page 11, lines 8-13. This sentence is confusing: it seems to imply that the increasing airborne fraction was due to changes in FAIR parameters (r_0, r_T, r_C, and c_j) *through time*. I rather assume that the increasing airborne fraction results from the structure of FAIR that leads to an ever-increasing \alpha, with changes FAIR parameters being responsible for the blue shading.

We have tried to make it clearer that the r_0, r_T and r_C are indeed constant over time and that the uncertainty corresponds to different values of these parameters in the revised manuscript.

39. Page 11, lines 14-23. FAIR results in Fig. 7d do not show a "straight-line relationship between cumulative carbon emissions and human-induced warming", the downward curvature being obvious starting from 500 GtC at least (not only at high cumulative emissions as mentioned). Please specify that the TCRE value provided is valid for 1000 GtC only.

We have revised the text to indicate the downward curvature is apparent across the range of cumulative emissions shown. We include a statement indicating that the value of TCRE is only exact for the first 1000GtC. Whilst recognising that there is some curvature to the relationship we choose to refer to it as 'approximately linear' to reflect the terminology widely used to refer to this simulated relationship in the literature.

40. Page 11, lines 24-26. This is Introduction-type text, not for the Results.

This text has been moved in the revised manuscript.

41. Page 11, line 28 to page 12, line 10. This text is not really related to FAIR and could easily be deleted. If not, it should be moved to a new subsection 3.5.

We believe that this section makes some important points that underlie our ranges for sampling TCR and RWF in the paper. We therefore keep this text as part of the new methods section.

42. Page 11, line 30. Gillett et al. (2013) highlighted the policy relevance of TCRE, not TCR.

We have altered this text (now in the methods section) to reflect that Gillett et al showed that thermal response uncertainty (therefore TCR) is the dominant uncertainty in the CMIP5 simulated range of TCRE.

43. Page 12, lines 10-16. The majority of this text belongs to 2.3, where explanations must be much improved. In particular, the c_j (not TCR and ECS) were changed in FAIR; how?

This text has been moved to the methods. See previous responses for our changes regarding how TCR and ECS are related to q_j.

44. Page 12, lines 16+. The text should refer to the results in Fig. 8b or this panel should be removed.

Panel 8b has now been removed from the revised manuscript.

45. Page 12, lines 17-23. The majority of this text belongs to 2.3, where the IPT equation and the distribution (shape and values) of d_1 must be justified. I also suggest combining what remains of this paragraph with the next one, as they are logically linked (i.e., there are no results for d_1 uncertainty only, right?).

We now include this text in the new methods section. We remove the IPT equation as our update the thermal response timescales (to reflect the mean to Geoffrey et al as opposed to the values used in IPCC-AR5) mean that is it no longer valid. Paragraphs are now combined in the revised manuscript.

46. Page 12, lines 24-27. The majority of this text belongs to 2.3, where the values chosen must be justified, the 300 random draws approach must be explained, and the "median" shown in Fig. 8 must be defined (is it the median of the 300 draws or the result obtained with the median values of the parameter distributions?).

We have moved this text and tried to justify our approach more in the revised manuscript.

47. Before the Conclusions. A short discussion of the following point is missing: the idea behind FAIR is to adjust the standard IRF time constant based on iIRF_100, which is by definition for a time horizon of 100 years. Do you expect FAIR to perform well for time horizons of 1000 years and more? Would this require expanding the time horizon of iIRF?

We include this discussion in the revised version of the manuscript as part of the conclusions, which felt a more natural fit for these points.

48. Page 13, line 10. Strictly speaking, the authors did not show that "including both explicit CO2 uptake- and temperature- induced feedbacks are essential". One could hypothesize that FAIR may give similar results working with temperature-induced feedback only but using a higher value of r_T (or with CO2 feedback only but using a higher value of r_C).

We would disagree with the reviewer on this point. We believe that we have shown the including temperature-induced carbon-cycle feedbacks are essential in order to replicate the 'radiatively-coupled' experiments under the prescribed concentration increase experiments. As mentioned in the text, the correct partitioning of the feedbacks will be important in scenarios with future differences in the relative contribution of CO2 and non-CO2 forcings on the climate-carbon-cycle system.

49. Page 15. The same study is referenced under both Meinshausen et al. 2011a and 2011b.

Corrected in the revised manuscript.

50. The Figures along with the related legends and captions are very poor. Here is only a subset of all the issues I saw.

- Fig. 1 caption: these results are for the historical period, so why the "RCP scenarios"?

We have updated the figure caption to give a more-complete statement on where

the non-CO2 forcings come from.

- Many panels lack the x-axis title.

All panels now have an x-axis title in the revised manuscript.

- There is often an overlap between axis scales (e.g., Fig. 1d bottom left) or with the panel letter (e.g., Fig. 5a bottom left).

Corrected in revised manuscript.

- Axes are not labelled consistently, for example "CO2 concentration" vs. "CO2 Concentrations" vs. "Atmospheric concentration", or "Airborne Fraction" vs. "Airborne fraction" vs. "Fraction of CO2 impulse remaining".

Corrected in revised manuscript.

- Results are often 'cut' because the scale is not large enough (e.g., Fig. 2c).

Corrected in revised manuscript.

- Why aren't MAGICC results shown on Figs. 2c and 2d; I presume they are available?

We have removed figure 2c and 2d from the revised manuscript.

- Fig. 2c shows results with very different scales, while Fig. 2d shows too many results. Fig. 2 should have 6 panels: a) as currently, b) as currently, c) temperature for RCP8.5, d) temperature for RCP2.6, e) temperature vs. emissions for RCP8.5, and f) temperature vs. emissions for RCP2.6.

We have removed figure 2c and 2d from the revised manuscript.

- Fig. 3 caption: the shading is grey, not black.

Corrected in revised manuscript.

- The legends of Fig. 6a and 6c should both appear in 6a as they apply to all panels of Fig. 6. The legend of Fig. 6b should be ordered logically.

Corrected in revised manuscript.

- Fig. 6 caption: replace "cumulative total carbon uptake" by "cumulative ocean and land carbon uptake".

Corrected in revised manuscript.

- There is no purple bar in Fig. 7a, in contradiction with both the legend and caption. There is no purple bar in Fig. 7d, in contradiction with the caption.

A purple bar was included in Figure 7a. We have altered the colour of the bar to aid visibility.

- Why aren't AR5-IR results shown in Figs. 7a and 7b?

AR5-IR results are not included in Figure 7a and 7b as these models share a thermal response model and these simulations are conducted with prescribed concentrations and not emissions, so would produce identical resutls. We have included a statement in the manuscript to make this point clear.

[revised manuscript text omitted]

---

## Author Response (AR3)

**Response to reviewers**

We respond to the remaining comments from reviewers in-line below...
A tracked changes version of the manuscript is appended to this document.

**Reviewer 1**

The manuscript describes a model which could be of use as a simplified emulator of ESMs. I agree that it is a step forward in comparison with a model which doesn't include the carbon feedback. A general comment: observed airborne CO2 fraction doesn't increase, although ESMs clearly project it. So there is something unaccounted in ESMs which obviously cannot be captured with a simple emulator as well. Beyond that, I have few minor comments listed below.

The title:
the global response of what? Temperature and airborne CO2 fraction? It should be specified in the title, there are many global characteristics of ESMs (eg., GPP, tree cover, ocean pH, marine export flux) which are not included in the impulse response model. The same is in the 2nd sentence in the abstract: "their range of response".

This has now been updated in the revised version of the paper to better reflect the variables of the model

Abstract:
p. 1. l. 5 idealized carbon-cycle experiments: be more precise with terminology. These are carbon-pulse and exponential CO2 increase experiments done with carbon cycle models.

Corrected in the text.

L 8. "A simple linear increase". Could it be non-simple linear increase? Linear is already simple, you do not need another word "simple".

"Simple" has now been removed.

l. 8-9. "a linear increase … is necessary and sufficient …". Sounds like you proved a mathematical theorem, but I missed both theorem and its proof in the paper. Conclusions have more careful language ("including feedbacks… are essential", p.15, l.17), ("the inclusion of feedback… offers an improvement", l.24-25), and even not strictly scientific expressions ("we believe that..., p.16, l. 11"). Could you adjust a confidence level of abstract to the one of conclusions?

Language adjusted in the revised version of the text.

p.1, l. 17: I disagree with the authors argumentation ".. such models are computationally intensive… ". For example, the MPI-ESM model I am working with in the T63 (CMIP5) resolution runs 70 years per day. There is no problem to run centennial-scale experiments with this model. The bottleneck is not the computational speed of ESMs, but the time needed for post-processing. Also, internal variability of ESMs requires either a strong forcing to get significant signal-to-noise ratio or to run the model many times in the ensemble mode. I think

the main reason to have a fast emulator of ESMs is a possibility to better explore uncertainty, such as done in probabilistic parameter sampling in your study.

These points have been expressed more clearly in the revised version of the text.

p. 4, eq. 3: what is j index staying for in Tj? Looking into the Table 1, j is either for deep or for upper ocean. On the other hand, T stays for surface temperature. Surface temperature of the deep ocean does not make much sense. I am confused here, please clarify.

We have added a clarification that the index j refers to the contribution to surface warming evolving on short/long timescales respectively.

**Reviewer 2**

The text and figures have been much improved in this revision of the paper. I found the discussion of the model equations and their relationship to physical reservoirs in the climate system to be clear and readable. The value of this model and its state-dependence was also clear, as was its limitations relative to more complex ESMs.

I found the discussion of normalization of the temperature response presented in Figure 3 to be unclear. I encourage the authors to better describe this normalization in the text.

We have clarified our normalisation of the temperature response in Figure 3c in the text.

I found the comparison in Figure 5 to be somewhat unconvincing. The authors plot results from their FAIR model relative to results from the pulse simulations from Herrington and Zickfield (2014). They discuss that these results showing a peak in the temperature within a deacde of emissions stopping were later shown to be misleading in a fully coupled model simulation (Zickfield and Herrington, 2015). It is therefore encouraging that the FAIR model does not necessarily simulate the near-term temperature peak, but it does not seem to be the relevant comparison. It would be better to see how the FAIR model response compares to the more complex model for a comparable pulse.

We have slightly redesigned Figure 5 to include the response of the UVic-ESCM from Herrington et al 2014 and expressed the y-axis as the fraction of maximum warming to more clearly illustrate the differences in the temporal evolution of warming in the AR5-IR, FAIR and UVic-ESCM under the pulse emission experiments of Herrington et al 2014.

[revised manuscript text omitted]